# Identification of gene regulatory networks affected across drug-resistant epilepsies

Liesbeth François [1,2,7] ✉, Alessia Romagnolo [2,7], Mark J. Luinenburg [2], Jasper J. Anink[2], Patrice Godard[1], Marek Rajman[1], Jonathan van Eyll [1], Angelika Mühlebner[2,3], Andrew Skelton[1], James D. Mills[2,4,5], Stefanie Dedeurwaerdere[1,8] & Eleonora Aronica [2,6,8] ✉

Epilepsy is a chronic and heterogenous disease characterized by recurrent unprovoked seizures, that are commonly resistant to antiseizure medications. This study applies a transcriptome network-based approach across epilepsies aiming to improve understanding of molecular disease pathobiology, recognize affected biological mechanisms and apply causal reasoning to identify therapeutic hypotheses. This study included the most common drug-resistant epilepsies (DREs), such as temporal lobe epilepsy with hippocampal sclerosis (TLE-HS), and mTOR pathway-related malformations of cortical development (mTORopathies). This systematic comparison characterized the global molecular signature of epilepsies, elucidating the key underlying mechanisms of disease pathology including neurotransmission and synaptic plasticity, brain extracellular matrix and energy metabolism. In addition, specific dysregulations in neuroinflammation and oligodendrocyte function were observed in TLE-HS and mTORopathies, respectively. The aforementioned mechanisms are proposed as molecular hallmarks of DRE with the identified upstream regulators offering opportunities for drug-target discovery and development.

Epilepsy is typically defined as a chronic disease characterized by recurrent unprovoked seizures[1]. However, the concept of epilepsy is evolving and it is recognized that besides seizures patients are also affected by cognitive, psychological, and social impairments[2,3], as well as increased mortality[4]. The heterogeneity in causes and clinical expression of the disease leads us to more commonly use the term epilepsies. There is an urgent need to identify therapeutic targets and develop tailored medications that go beyond the current anti-seizure medications (ASMs)[5], both in efficacy and in addressing the disease starting from the pathobiology. Discriminating the factors contributing to different subtypes of drug-resistant epilepsy (DRE) would shed light on the pathobiological mechanisms that are shared or specific across disease types, and enable hypotheses to be

established for developing precision medicines to ensure better patient care.

Here, we focused on some of the most common forms of DREs, temporal lobe epilepsy with hippocampal sclerosis (TLE-HS) and malformations of cortical development, including focal cortical dysplasia type IIa and type IIb (FCD IIa and FCD IIb) and cortical tubers in tuberous sclerosis complex (TSC). TLE-HS is characterized by selective neuronal cell loss with concomitant astrogliosis in the hippocampus[6]. FCD type II and TSC cortical tubers are characterized by hyperactivation of the mTOR-signaling pathway and collectively termed mTORopathies[7]. Furthermore, both pathologies are characterized by common histopathological hallmarks such as cortical dyslamination, dysmorphic neurons, and large immature cells called balloon cells in

[1]UCB Pharma, Early Solutions, Braine-l'Alleud, Belgium. [2]Department of (Neuro)Pathology, Amsterdam UMC, University of Amsterdam, Amsterdam Neuroscience, Amsterdam, The Netherlands. [3]Department of Pathology, University Medical Center Utrecht, Utrecht, The Netherlands. [4]Department of Clinical and Experimental Epilepsy, UCL Queen Square Institute of Neurology, London, UK. [5]Chalfont Centre for Epilepsy, Chalfont St Peter, Chalfont, UK. [6]Stichting Epilepsie Instellingen Nederland (SEIN), Heemstede, The Netherlands. [7]These authors contributed equally: Liesbeth François, Alessia Romagnolo. [8]These authors jointly supervised this work: Stefanie Dedeurwaerdere, Eleonora Aronica. ✉e-mail: liesbeth.francois@ucb.com; e.aronica@amsterdammumc.nl

FCD IIb (absent in FCD IIa) or giant cells in TSC cortical tubers[8,9]. Despite the large research efforts to elucidate the molecular mechanisms underlying epilepsies, the molecular profile contributing to the epileptogenicity in TLE-HS and the mTORopathies is not completely understood.

Discovering disease pathways has the potential to reveal druggable targets that could restore impaired gene expression back to homeostasis. The network-based system analysis Causal Reasoning Analytical Framework for Target discovery (CRAFT) previously identified epilepsy-specific gene coexpression modules (i.e. sets of coexpressed genes) in a pilocarpine mouse model, allowing the identification of therapeutic candidates[10]. Here, gene coexpression modules allowed for the assembly of an unbiased, global model of the pathobiology based on the assumption that biological pathways are dysregulated in the disease state. CRAFT identifies potential upstream regulators by predicting the interaction between cell membrane receptor proteins (CMPs), transcription factors (TFs), and downstream target genes[10].

To our knowledge, available transcriptomics datasets for epilepsy are often limited to one pathology, lacking comparison across epilepsies, and are low in sample number[11–17]. Therefore, further investigation of a larger cohort involving different pathologies can extend our understanding of the pathobiological mechanisms that underly epilepsy.

This study enabled the construction of the global molecular signature of epilepsies by comparing disease transcriptional profiles, and identified key underlying mechanisms shared across epilepsies that are involved in neurotransmission and synaptic plasticity, immune response, brain extracellular matrix (ECM), and energy metabolism. In addition, specific dysregulations in neuroinflammation and neuronal support, and myelination were identified in TLE-HS and mTORopathies, respectively. We propose that these mechanisms are the putative molecular hallmarks of DRE and may be active players in disease progression. The upstream regulators identified here by causal reasoning offer hypotheses to test their effect on disease and, potentially, generate opportunities for drug-target discovery.

## Results

This study provided a data-driven framework for the systematic identification of dysregulated biological pathways in the disease state and to categorize global epilepsy mechanisms across DREs. The identification of impaired transcriptional coregulations in and across different epilepsy pathologies combined with predicted mechanistic regulatory hypotheses can be leveraged experimentally to test their therapeutic potential.

### Transcriptional differentiation between cohorts by tissue type and disease

In total, 28,366 expressed genes (mapped reads ≥6 counts in at least 20% of samples within each cohort) were detected across the cohorts. First, to obtain a global understanding of the transcriptional landscape and assess potential differentiation between clinical cohorts, sample clustering was explored using both unsupervised hierarchical clustering and supervised discriminant analysis on principal components to identify discriminatory features between cohorts.

The unsupervised hierarchical clustering showed that the TLE-HS cohort could be distinguished from the mTORopathies cohort, and further, there was no clear separation within the latter (Fig. 1a). Discriminant features associated with tissue on the first component (cortex vs hippocampus) and disease status on the second component (epilepsy vs healthy) were identified (Fig. 1b, c). However, as the epilepsy condition is partly defined by the brain area of seizures origin, the effect of tissue and disease could not be assessed independently. Figure 1d shows the prior and posterior assignment of individuals to

the cohorts which indicated a good reassignment rate for TLE-HS. A lower reassignment rate for the mTORopathies, specifically for FCD IIa patient samples, where only half of the individuals were reassigned to their cohort (Fig. 1d), indicated difficulty in discriminating between these populations when taking all six cohorts together.

A focused analysis was performed on the three mTORopathies cohorts to explore their transcriptional similarity. The first discriminant component and reassignment proportion suggest a gradual change in gene expression profile in individuals diagnosed with FCD IIa that were reassigned to FCD IIb but not TSC (Fig. 1e). Similarly, more overlap was found between TSC and FCD IIb than with FCD IIa (Fig. 1e). Based on these results, all three pathologies will be considered as an additional meta-cohort to explore potential shared regulations between mTORopathies.

### Identification of gene coexpression modules within epilepsy pathologies

It is hypothesized that gene coexpression modules ('gene modules') can build an unbiased, global model of epilepsy pathobiology based on the assumption that some biological pathways may be differentially regulated in the disease state due to perturbations of gene expression control. The workflow to annotate the identified gene modules is described in the Materials and "Methods" section (Fig. 4). Briefly, pathway and cell-type annotations aimed to unravel the underlying pathobiology of the diseases. Furthermore, the differential coexpression of gene modules between disease and healthy control samples brought to light the gene modules impacted in the disease state. Finally, the correlation of each gene within each module is assumed to be the consequence of a common (set of) upstream transcriptional regulator(s) activity. The causal reasoning framework, CRAFT, predicts upstream regulators (transcriptional regulators, TFs, and miRNA, as well as CMPs) that, based on current knowledge, could affect the modules to form an actionable regulatory hypothesis.

This workflow was applied to all cohorts (TLE-HS, FCD IIa, FCD IIb, and TSC) except the FCD IIa cohort due to insufficient sample numbers. Figure 2 shows the change in gene coexpression ($R^2$) highlighting the annotated biology for the affected modules related to multiple brain functions such as neurotransmission and synaptic plasticity, immune response, and energy metabolism among others. No association to phenotype and antiseizure medications was identified for the modules in any cohort. A summary of the results of the identified gene modules per cohort is described in Table 1. The next paragraphs describe the most affected gene modules and there are further details in Supplementary Data 1–4.

#### TLE-HS

For TLE-HS, 37 gene modules were identified with nine modules presenting a significant change in coexpression as measured by $R^2$ between disease and healthy control patient samples, indicating that these modules were significantly affected in TLE-HS (Fig. 2a, panel TLE-HS) (Supplementary Data 2). For example, TLE.13.o, TLE.7.o and TLE.12.u were the most perturbed modules with more than 50 genes per module with an $\Delta R^2$ ranging between 0.24 and 0.32 (Supplementary Data 1). These modules highlighted different biological function as affected in epilepsy (immune response/neuroinflammation, extracellular matrix function, and mRNA/protein processing) (Supplementary Data 3 and 4) Multiple upstream regulators were identified using the causal reasoning framework. For TLE.13.o up to 26 module regulators were predicted, including miRNAs (2), TF (14), and CMPs (328) (Supplementary Data 1). For TLE.7.o up to 366 regulators were predicted, including TF (4) and CMP (275) with no candidate regulators for TLE.12.u (Supplementary Data 1). Overall, out of the nine gene modules identified to be affected in epilepsy, transcriptional regulators and CMPs were available for six and four gene modules, respectively.

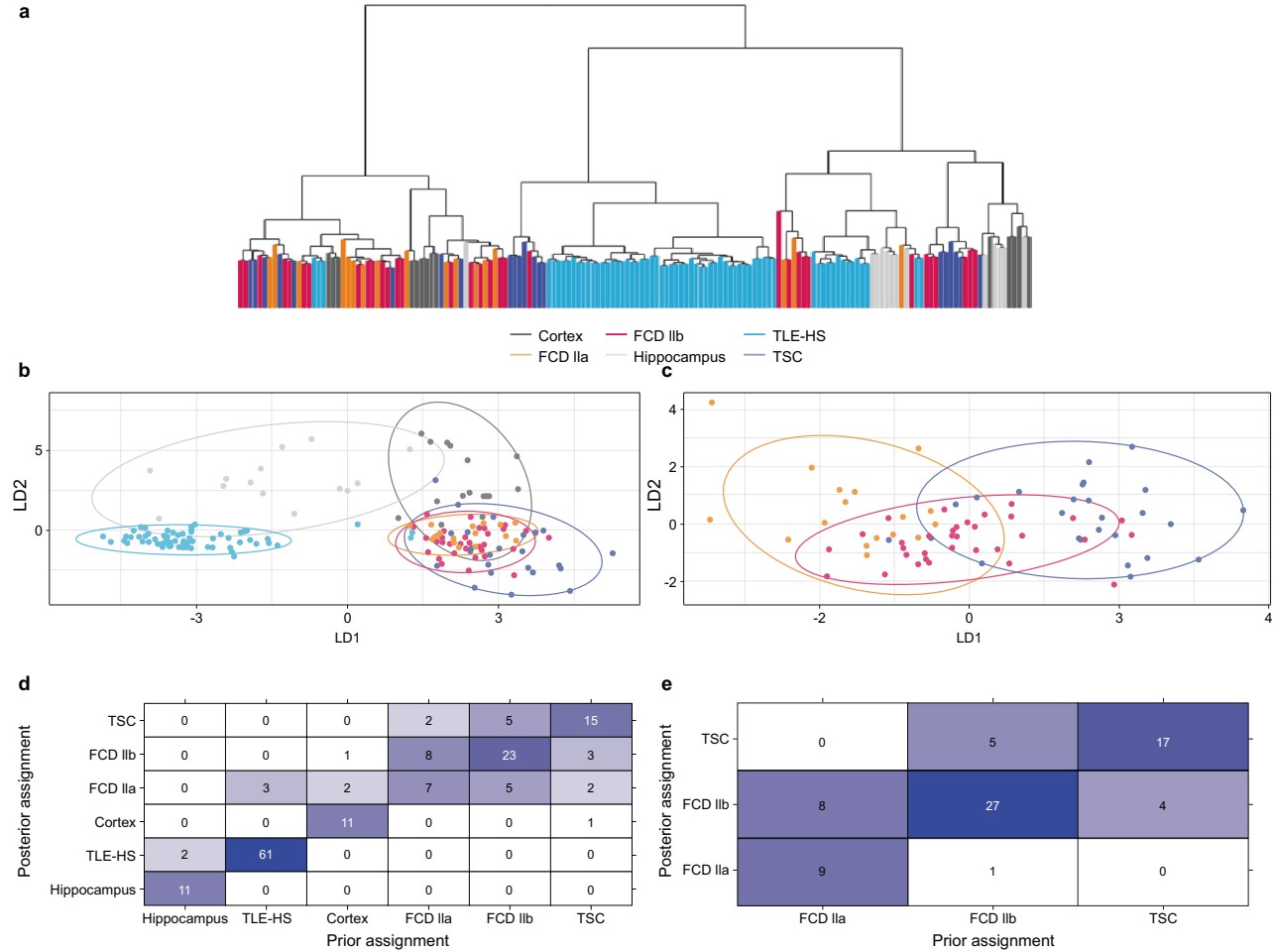

**Fig. 1 | Comparison of transcriptional profile across cohorts. a** Dendrogram based on unsupervised hierarchical clustering including all epilepsy (TLE-HS, FCD IIa, FCD IIb, and TSC) and control (cortex and hippocampus) patient samples. **b** Discriminant analysis on principal components on all cohorts identified discriminating features by tissue on the first component (linear discriminant 1 – LD1) and disease status on the second component (linear discriminant 2 – LD2). **c** Discriminant analysis on principal components on mTORopathy cohorts only (FCD IIa, FCD IIb, and TSC) identified limited separation on the first discriminant function. **d** Prior and posterior cohort assignment after discriminant analysis on principal components on all cohorts. The prior and posterior assignment of individuals to the cohort based on the discriminant functions is provided indicating

admixture between cohorts. The numbers in the heatmap indicate how many samples of each cohort are (re)assigned to the same cohort based on the discriminant functions. **e**, Prior and posterior cohort assignment after discriminant analysis on principal components on mTORopathies specifically. The prior and posterior assignment of individuals to the cohort based on the discriminant functions were provided indicating admixture between cohorts. The numbers in the heatmap indicate how many samples of each cohort are (re)assigned to the same cohort based on the discriminant functions. FCD focal cortical dysplasia, TLE-HS temporal lobe epilepsy with hippocampal sclerosis, TSC tuberous sclerosis complex. Source data are provided as a Source Data file.

## FCD IIb

The analysis of FCD IIb identified 28 gene modules with 22 gene modules significantly differentially coexpressed (Fig. 2a, panel FCD IIb) (Supplementary Data 2). Gene modules that showed significant differential coexpression were involved in immune response, oligodendrocyte function, oxidative phosphorylation among others (Supplementary Data 3 and 4). The most affected modules FCD2b.7.o and FCD2b.14.u ($\Delta R^2$ ranging between 0.49 and 0.54) captured less than 20 genes, limiting their relevance (Supplementary Data 1). Modules FCD2b.5.o, FCD2b.6.o, and FCD2b.6.u contained between 240 and 330 genes with functions related to mRNA translation (FCD2b.5.o), oxidative phosphorylation (FCD2b.6.o) and endosome function (FCD2b.6.u) (Supplementary Data 4). Overall, six of the 28 identified gene modules lacked functional annotation. The causal reasoning identified multiple regulatory hypotheses. For FCD2b.5.o, one TF (SAFB) and 19 upstream CMPs were predicted (Supplementary Data 1). For FCD2b.6.o, 62 transcriptional regulators (60 miRNA/2 TF) and 33 upstream CMPs were predicted. No upstream regulator could be identified for FCD2b.6.u (Supplementary Data 1).

## TSC

In TSC, 30 gene modules were identified with 23 gene modules significantly differentially coexpressed (Fig. 2a, panel TSC) (Supplementary Data 2). The strongest differential coexpression resulted for modules TSC.11.u, TSC.13.o, TSC.13.u, and TSC.14.o containing 120–290 genes in the modules with a $\Delta R^2$ ranging from 0.48 to 0.52 (Supplementary Data 1). These four modules were enriched for a broad spectrum of different functions, such as modulation of chemical synaptic transmission, positive regulation of cytokine production, postsynaptic density, and interferon signaling (Supplementary Data 4). Like FCD IIb, not all affected modules could be biologically annotated despite utilizing different pathway resources (Supplementary Data 4). CRAFT identified two TFs as well as 12 CMPs for TSC.11.u (Supplementary Data 1). For TSC.13.o, 21 transcriptional regulators (3 miRNA / 18 TF) and 380 upstream CMPs were found. Although no upstream regulators were identified for TSC.13.u, 68 transcriptional regulators were predicted for TSC.14.o (2 miRNA / 66 TF) as well as 392 upstream CMPs (Supplementary Data 1).

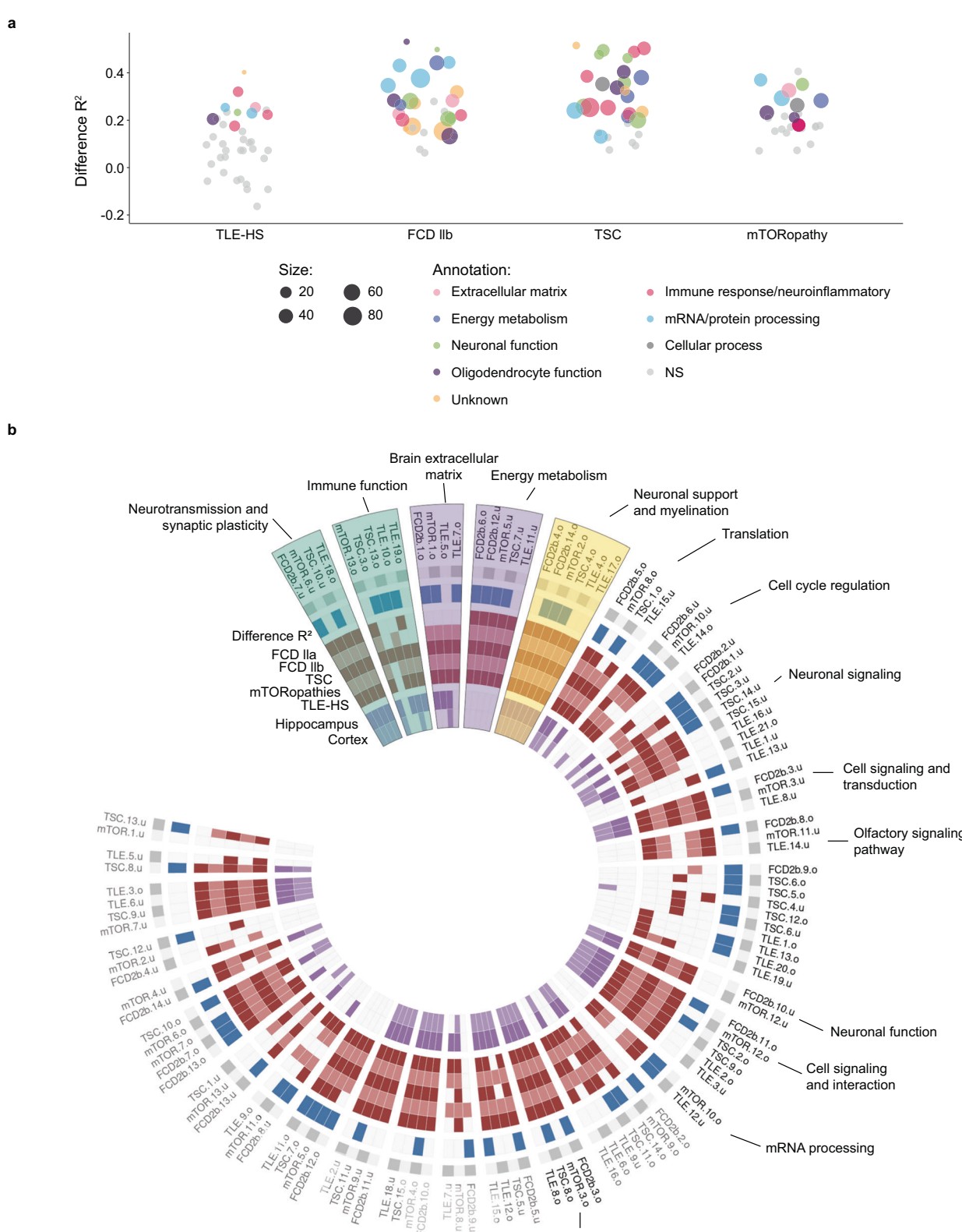

## mTORopathies

In the mTOR cohort (all FCD IIa, FCD IIb, and TSC samples), 28 gene modules were identified but only nine gene modules were found differentially coexpressed (Fig. 2a, panel mTORopathy) (Supplementary Data 2). The strongest significant differential coexpression could be identified for gene modules mTOR.1.o (393 genes), mTOR.10.o (293 genes), mTOR.10.u (257 genes), and mTOR.1.o (3 genes) with R² ranging

from 0.33 to 0.35 (Supplementary Data 1). Due to the limited size of mTOR.1.u, only the remaining three modules will be described further here. Functional annotation of these modules related to RNA splicing, response to topologically incorrect protein folding, and extracellular matrix organization (Supplementary Data 4). CRAFT could not identify any upstream regulators for gene module mTOR.10.o, whereas for mTOR.1.o it identified 41 potential transcriptional regulators (4 miRNA /

**Fig. 2 | Overview of the gene modules per epilepsy cohorts (TLE-HS, FCD IIb, TSC, and mTORopathies). a** Overall comparison of the different gene modules indicating the change in $R^2$ between epilepsy patient samples and healthy control samples for each analyzed epilepsy cohort. Gene modules were annotated when differentially coexpressed by their main inferred biological function. **b** Circular heatmap showing identified regulomes derived from the systematic comparison of all identified modules by the different metrics. From outside to the inside: the gene module names were shown, the effect on disease based on differential $R^2$ (blue), conservation in epilepsy cohorts (red), and conservation in healthy control (purple). Labels of regulomes lacking functional annotation were colored in gray, regulomes with consistent functional annotation were labeled in black. The highlighted regulomes in blue, purple, and yellow represent the 'enhanced', 'activated', and 'pathology-specific' regulomes, respectively, that were selected. FCD focal cortical dysplasia, TLE-HS temporal lobe epilepsy with hippocampal sclerosis, TSC tuberous sclerosis complex. Source data are provided as a Source Data file.

**Table 1 | Summary table of gene module identification, annotation, and causal reasoning predictions within each epilepsy cohort**

| Pathology | Module genes[a] | Modules[b] | DC[c] | Functional annotation[d] | CRAFT (TF/CMP)[e] | TF/miRNA[f] | CMP[g] |
|---|---|---|---|---|---|---|---|
| TLE-HS | 4481 | 37 | 9 | 28 | 20/17 | 1581 | 508 |
| FCD IIb | 9928 | 28 | 22 | 24 | 21/17 | 918 | 456 |
| TSC | 9453 | 30 | 23 | 26 | 17/17 | 1051 | 489 |
| mTOR | 7466 | 26 | 9 | 23 | 16/14 | 1069 | 463 |

*CMP* cell membrane receptor protein, *FCD* focal cortical dysplasia, *miRNA* microRNA, *mTOR* pathway-related malformations of cortical development, *TF* transcription factor, *TLE-HS* temporal lobe epilepsy with hippocampal sclerosis, *TSC* tuberous sclerosis complex.
[a]Number of genes assigned to modules.
[b]Number of identified modules.
[c]Number of significantly differentially coexpressed modules per analysis.
[d]Number of modules for which functional annotation is available.
[e]Number of modules for which a direct TF or indirect CMP is available.
[f]Number of predicted transcriptional regulators, including both TFs and miRNA.
[g]Number of predicted CMPs.

37 TF) and 384 upstream CMPs. Similarly, for mTOR.10.u, 51 transcriptional regulators (49 miRNA / 2 TF) and 25 upstream CMPs were identified (Supplementary Data 1).

Identified affected gene module and regulators may provide opportunities to modulate these networks and restore their homeostatic gene expression profile. Figure 2a shows the identification of neurotransmission and synaptic plasticity, immune response, brain ECM, energy metabolism, neuronal support, and myelination affected in epilepsy. To enable a global understanding of the regulation of pathobiology of epilepsy, the next section discusses the overall comparison of these identified modules and their regulators.

## Connecting gene modules across epilepsy cohorts identifies shared biology

The gene coexpression module analysis identified modules related to similar biological functions across the different epilepsy patient cohorts. Here, systematic comparison based on all identified modules was performed to enable a global and objective understanding of conserved or disease-specific modules. Unsupervised clustering of gene modules based on the inclusion index identified clusters of gene modules that were functionally annotated to infer their potential shared biology. These clusters are termed regulomes to better capture the functional role of cluster of gene modules as global regulatory pathways in the epilepsy pathobiology. In this context, a regulome refers to the transcriptional regulation that may depend on the pathological state of the tissue[18]. Finally, the shared predicted TFs by the individual CRAFT analyses were listed as candidate regulators with potential to act across epilepsies.

Differential coexpression and conservation was used to measure activity states across the different pathologies enabling the regulomes to be separated into four different categories: constitutive, enhanced, activated, and pathology-specific regulomes. Constitutive regulomes show no change between the control and epilepsy patient samples. Enhanced regulomes are conserved in cohorts but showed significant increased activity in epilepsy. Activated regulomes are only present and active in epilepsy. Finally, some gene modules did not present a strong overlap with gene modules from any other epilepsy conditions; however, as these modules were differentially coexpressed in a specific epilepsy cohort, these were referred to as pathology-specific regulomes.

The analysis revealed 29 regulomes total varying in size from two to 10 gene modules (Fig. 2b, Supplementary Data 5). Here, regulomes ($n = 14$) with a consistent functional annotation across multiple pathway databases and effect in epilepsy were identified and selected (Fig. 2b). Based on the classification described above, regulomes related to neurotransmission and synaptic plasticity, immune response, brain ECM, energy metabolism and oligodendrocyte function are highlighted (Fig. 2b).

### Immune response and neuroinflammation

The discrimination between clusters enriched for immune response pathways and neuroinflammation relies on the pathway annotations. Neuroinflammation concerns the process mediated by resident central nervous system glia (microglia and astrocytes) and endothelial cells[19], whereas immune response is defined as the reaction of the body against the impaired homeostasis involving the recruitment of immune cells leading to a systemic response[19]. Although regulomes can show a stronger association with one or another, differentiation between immune response and neuroinflammation regulomes is not absolute and they are presented here together.

The first regulome enriched for immune response and neuroinflammation belongs to the enhanced regulomes capturing modules TLE.10.o, TLE.19.o, TSC.3.o, TSC.13.o, and mTOR.13.o (Fig. 2b, Supplementary Fig. 1). The enrichment for the intersecting genes showed enrichment for immune response; antigen presentation by MHC class I: cross-presentation (MetaBase), neutrophil degranulation (Reactome), positive regulation of cell activation and immunoglobulin binding (GO) (Supplementary Data 4 and 5). Cell-type marker enrichment from PanglaoDB identified significant overlap with markers from macrophages and microglia (Supplementary Data 4 and 5). These immune response-related gene modules showed a differentiated effect across the different cohorts, with significant increase in gene coexpression detected in TLE (TLE.10.o) and TSC (TSC.3.o and TSC.13.o). In contrast, module TLE.19.o and mTOR.13.o showed no activation in the TLE-HS and mTORopathy cohorts (Supplementary Fig. 1a). Conservation statistics also differed between the cohorts. For

TLE-HS the regulome was conserved in hippocampus controls but not in cortex controls. Similarly, module TLE.19.o was not conserved in FCD IIb whereas module TLE.10.o was not conserved in either FCD IIa or IIb. The TSC modules showed no conservation of coexpression in control cortex indicating the activated status of this particular regulome in the disease state, in alignment with the strong observed differential coexpression. mTOR.13.o showed conservation in control and all epilepsy cohorts but, similarly, no change in coexpression comparing disease and control cohorts (Supplementary Data 3). Finally, several common transcriptional regulators, such as SPI1, ETS1, STAT1, IRF8, and NF-κB were consistently predicted to activate their downstream genes, with the single exception of STAT3 which showed inhibition of module TSC.3.o and mTOR.13.o while activating modules mTLE.10.o, mTLE.19.o, and TSC.13.o (Supplementary Fig. 1b).

A pathology-specific regulome (module TLE.20.o) was identified related to immune response; IL-1 signaling pathway and innate immune response to contact allergens (MetaBase), interleukin-4 and interleukin-13 signaling and interleukin-10 signaling (Reactome) and inflammatory response (GO) (Supplementary Data 4 and 5). In addition, this gene module was enriched for cell-type markers related to microglia (PanglaoDB) (Supplementary Data 4 and 5). Although several gene modules across different cohorts were related to microglia function, TLE.20.o has a limited gene overlap with any of the other

identified gene modules in the FCD IIb, mTOR or TSC cohorts (Supplementary Data 1, 4, and 5). This specific module showed a stronger and significant coregulation in brain tissues from TLE patients versus control post-mortem samples (Supplementary Fig. 1c).

### Neuronal support and myelination

The neuronal support and myelination regulome includes FCD2b.4.o, FCD2b.14.o, mTOR.2.o, TSC.4.o, TLE.4.o, and TLE.17.o (Fig. 2b, Supplementary Fig. 1). However, only mTORopathies gene modules FCD2b.14.o, mTOR.2.o and TSC.4.o were significantly perturbed, except FCD2b.4.o and TLE-HS modules, TLE.4.o and TLE.17.o (Fig. 3a). Therefore, this neuronal support and myelination regulome was assigned as pathology-specific. The following annotations triacylglycerol metabolism (MetaBase), G alpha (i) signaling events (Reactome), ensheathment of neurons and actin binding (GO) were identified as enriched in each module (Supplementary Data 4 and 5). In addition, the intersecting genes showed significant overlap with oligodendrocyte cell-type markers (PanglaoDB) (Supplementary Data 4 and 5). The regulations of all gene modules were conserved in both control and disease samples but enhanced in the disease state. The two most common upstream transcriptional regulators identified by CRAFT were SOX10 which activated the modules and miR-488-5p which inhibited the expression of genes belonging to the gene modules

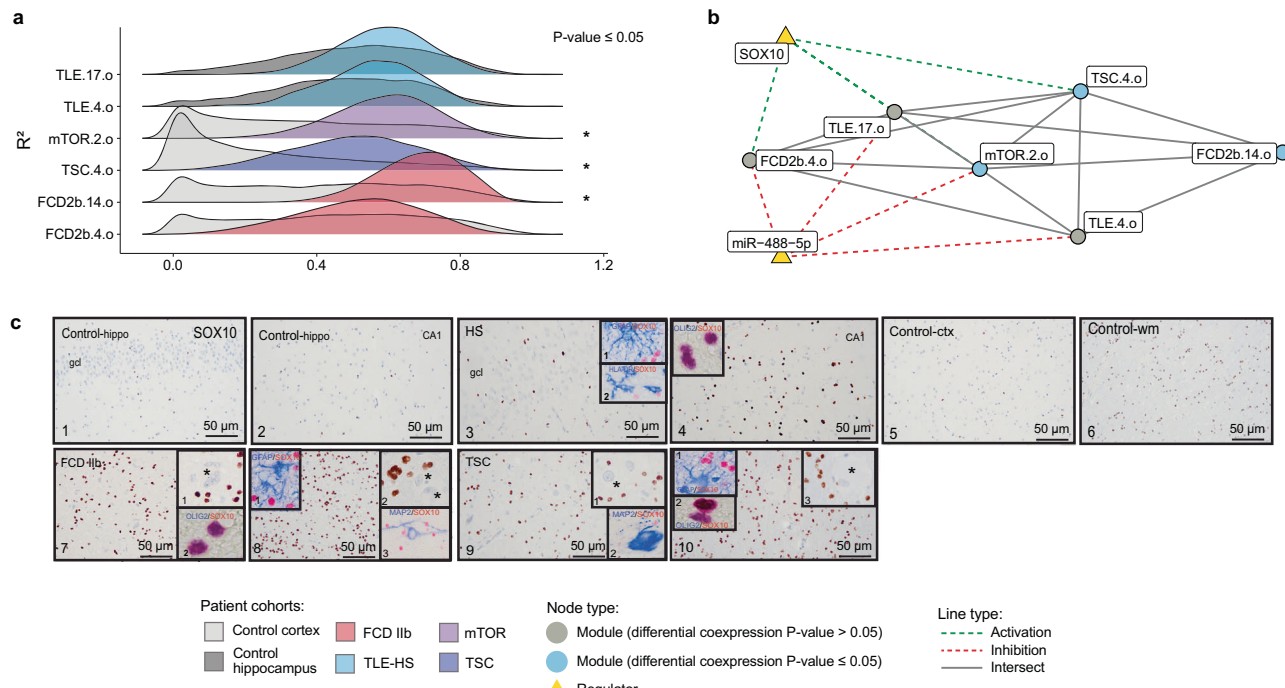

**Fig. 3 | Gene modules differential coexpression for multiple regulomes related to pathological mechanisms.** Network showing the gene overlap size between different gene modules and upstream transcriptional regulators. Cellular expression pattern of SOX10 immunoreactivity (IR) assessed in TLE-HS, FCD IIb, and TSC ($n = 3$ biological replicates per cohorts, $n = 2$ technical replicates). **a** The ridgeplots showed the distribution of gene modules coexpression ($R^2$) for epilepsy and control patient cohorts within neuronal support and myelination regulome. Statistical significance of differential coexpression was assessed using a two-sided permutation test (mTOR.2.o $p$-value $= 3.6 \times 10^{-2}$, TSC.4.o $p$-value $= 1.1 \times 10^{-2}$, FCD2b.14.o $p$-value $= 2.09 \times 10^{-2}$). **b** Neuronal support and myelination network with indication of differential coexpression of the relevant gene modules. SOX10 and miR-488-5p were predicted as common transcriptional regulators showing activation or inhibition effect on the gene modules. **c** Cellular expression pattern of SOX10 immunoreactivity (IR) in hippocampal sclerosis (HS), focal cortical dysplasia type IIb (FCD IIb), and tuberous sclerosis complex (TSC). Panels 1–2 (control hippocampus; gcl, granule cell layer) and panels 3–4 (hippocampal sclerosis, HS): nuclear expression of SOX10 was restricted to oligodendroglial cells; insert 1 in panel 3:

SOX10 (red) was not detectable in GFAP (blue) positive cells (astrocytes); insert 2 in panel 3: SOX10 (red) was not detectable in HLA-DR (blue) positive cells (microglial cells); insert in panel 4: SOX10 (red) co-localized with OLIG2 (blue) positive cells. Panels 5–6 (control cortex, 5 and white matter, 6), panels 7–8 (FCD IIb), and panels 9–10 (TSC): nuclear expression of SOX10 was restricted to oligodendroglial cells; insert 1 in panels 7–8: SOX10 positive cells surrounding negative balloon cells (asterisks). Insert 2 in panel 7: SOX10 (red) co-localized with OLIG2 (blue) positive cells; insert 1 in panel 8: SOX10 (red) was not detectable in GFAP (blue) positive cells; insert 3 in panel 8: SOX10 (red) was not detectable in MAP2 (blue) positive cells. Insert 1 in panels 9 and 10: SOX10 positive cells surrounding a negative dysmorphic neuron (asterisk in 1 in panel 9) and a negative giant cell (asterisk in 3 in panel 10); insert 2 in panel 10: SOX10 (red) co-localized with OLIG2 (blue) positive cells. Scale bars: 50 μm. FCD focal cortical dysplasia, GFAP glial fibrillary acidic protein, HLA human leukocyte antigen, TLE-HS temporal lobe epilepsy with hippocampal sclerosis, TSC tuberous sclerosis complex. Source data are provided as a Source Data file.

(Fig. 3b). The cellular expression pattern of SOX10 immunoreactivity (IR) was confirmed in oligodendroglial cells in TLE-HS, FCD IIb, and TSC samples (Fig. 3c).

## Brain extracellular matrix

Modules FCD2b.1.o, mTOR.1.o, mTLE.5.o, and mTLE.7.o were identified in brain ECM-activated regulome. Significant enrichment was found for cytoskeleton remodeling (MetaBase), extracellular matrix organization (Reactome), supramolecular fiber organization, and extracellular matrix structural constituent (GO), as well as enrichment for markers of Bergmann glia, the highly specialized radial astrocytes of the cerebellar cortex (PanglaoDB) (Supplementary Data 4 and 5). Among the gene modules involved in this regulation, mTOR.1.o, mTLE.7.o, and FCD2b.1.o showed a significant increase in coexpression (Fig. 4a). This regulome was not conserved in control patient samples but became activated in the disease cohorts (Fig. 4a). Finally, a common transcriptional regulator was identified to activate regulation of modules, namely SP1 (Fig. 4b). The cellular expression pattern of SP1 immunoreactivity (IR) was confirmed in astroglial cells in TLE-HS samples, whereas control hippocampus only showed low expression of SP1 in neuronal cells (Fig. 4c). Similarly, in control cortex the expression of SP1 was low in neuronal cells and sporadic in astrocytes within the white matter. In FCD IIb and TSC, SP1 IR was observed in dysplastic neurons, astrocytes, and balloon cells/giant cells, whereas microglia/macrophages showed absence of SP1 expression.

## Energy metabolism

The regulome capturing energy metabolism consists of FCD2b.6.o, mTOR.5.u, TSC.7.u, FCD2b.12.u, and mTLE.11.o (Fig. 2b). As this regulome was affected in the epilepsy cohort only, it was classified as activated. Functional annotation associated with this module included oxidative phosphorylation (MetaBase), respiratory electron transport (Reactome), and generation of precursor metabolites and energy (GO). However, no annotation with cell-type markers from PanglaoDB could be identified (Supplementary Data 4). All gene modules showed an increase in coexpression but significance was only reached for gene modules FCD2b.6.o, mTOR.5.o, TSC.7.u, and FCD2b.12.u (Fig. 5a). None of these gene modules were conserved in the control cohorts (Fig. 5a). The most common transcriptional regulator KMD1A (LSD1) was predicted to activate gene modules FCD2b.12.u, TSC.7.u, and mTOR.5.u (Fig. 5b). Cellular expression patterns of KDM1A IR in TLE-HS, FCD IIb, and TSC (Fig. 5c) showed restricted neuronal expression in control hippocampus, contrary to nuclear expression in both neurons and astrocytes in TLE-HS resected hippocampus. Similarly, in control cortex and white matter, the expression of KDM1A was restricted to neuronal cells, whereas FCD IIb and TSC showed KDM1A expression in dysplastic neurons, astrocytes and balloon cells/giant cells (Fig. 5c). As the IHC of epilepsy cohorts showed a consistent expression of KDM1A in astrocytes, in vitro validation of the role of KDM1A was assessed using PMA/Ionomycin stimulated

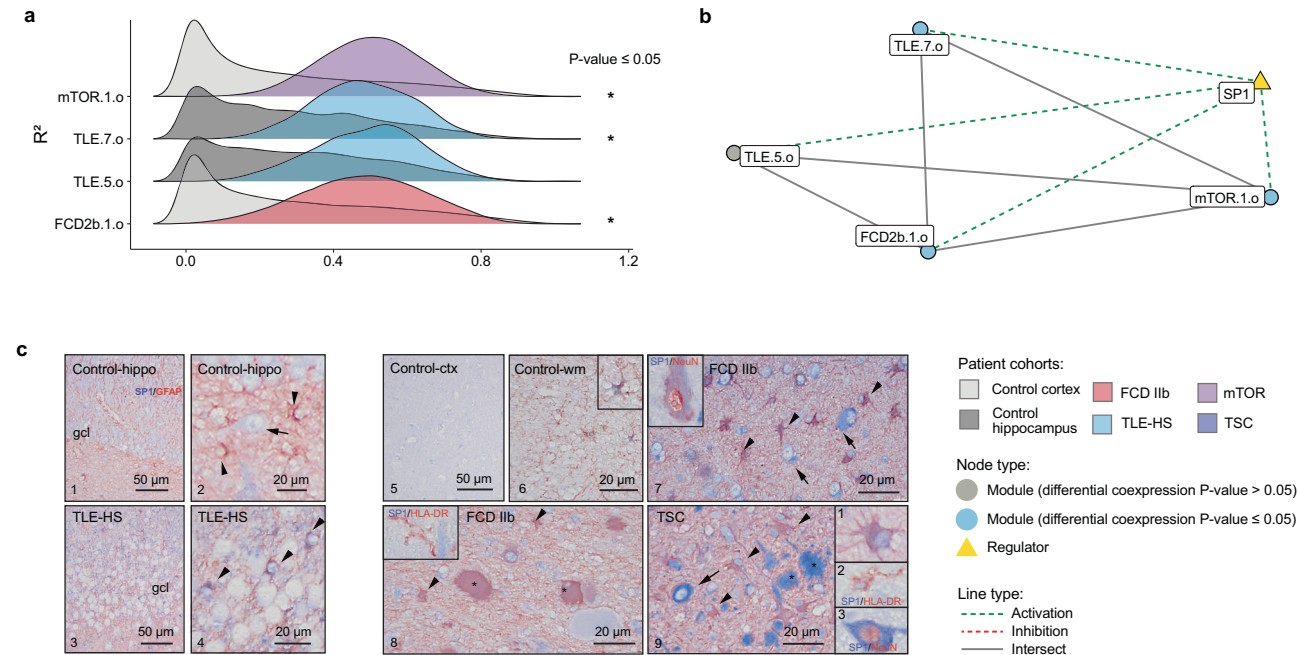

**Fig. 4 | Gene modules differential coexpression for multiple regulomes related to pathological mechanisms.** Network showing the gene overlap size between different gene modules and upstream transcriptional regulators. Cellular expression pattern of SP1 immunoreactivity (IR) assessed in TLE-HS, FCD IIb, and TSC ($n = 3$ biological replicates per cohorts, $n = 2$ technical replicates). **a** The ridgeplots showed the distribution of gene modules coexpression ($R^2$) for epilepsy and control cohorts within brain extracellular matrix regulome; mTOR.1.o, TLE.7.o, and FCD2b.1.o gene modules showed a significant increase of $R^2$. Statistical significance of differential coexpression was assessed using a two-sided permutation test (TSC.13.0 $p$-value = $9.99 \times 10^{-4}$, TSC.3.0 $p$-value = $4.00 \times 10^{-3}$, TLE.10.o $p$-value = $3.28 \times 10^{-2}$). **b** Brain extracellular matrix network highlighting the differentially coexpressed gene modules. SP1 was predicted as a common transcriptional regulator showing activation effect on the gene modules. **c** The cellular expression pattern of SP1 IR was assessed in TLE-HS, FCD IIb, and TSC. Panels 1–9: IHC of SP1. Panels 1–2 In control hippocampus, SP1 expression was very low in neuronal cells (arrow in panel 2, hilar

neuron); SP1 was not detectable in GFAP-positive cells. Panels 2–4: In TLE-HS, SP1 expression in astroglial cells (arrowheads). Panels 5–6: In control cortex, very low expression of SP1 (panel 5); occasionally few GFAP-positive cells were observed in the white matter (wm) (panel 6). Panels 7–8: In FCD IIb, SP1 IR was observed in dysplastic neurons (arrows) and GFAP-positive cells (arrowheads), including GFAP-positive balloon cells (asterisks). SP1 expression in a NeuN dysplastic neuron (insert in panel 7). Absence of SP1 expression in HLA-DR positive cells (microglia/macrophages; insert in panel 8). Panel 9: In TSC, SP1 expression in dysplastic neurons (arrow; high-magnification of a dysplastic neuron; insert 3 in panel 9) and GFAP-positive cells (arrowheads; insert 1 in panel 9), including giant cells (asterisks). Absence of SP1 expression in HLA-DR positive cells (microglia/macrophages; insert 2 in panel 9). Scale bars: 50 μm. FCD focal cortical dysplasia, GFAP glial fibrillary acidic protein, HLA human leukocyte antigen, TLE-HS temporal lobe epilepsy with hippocampal sclerosis, TSC tuberous sclerosis complex. Source data are provided as a Source Data file.

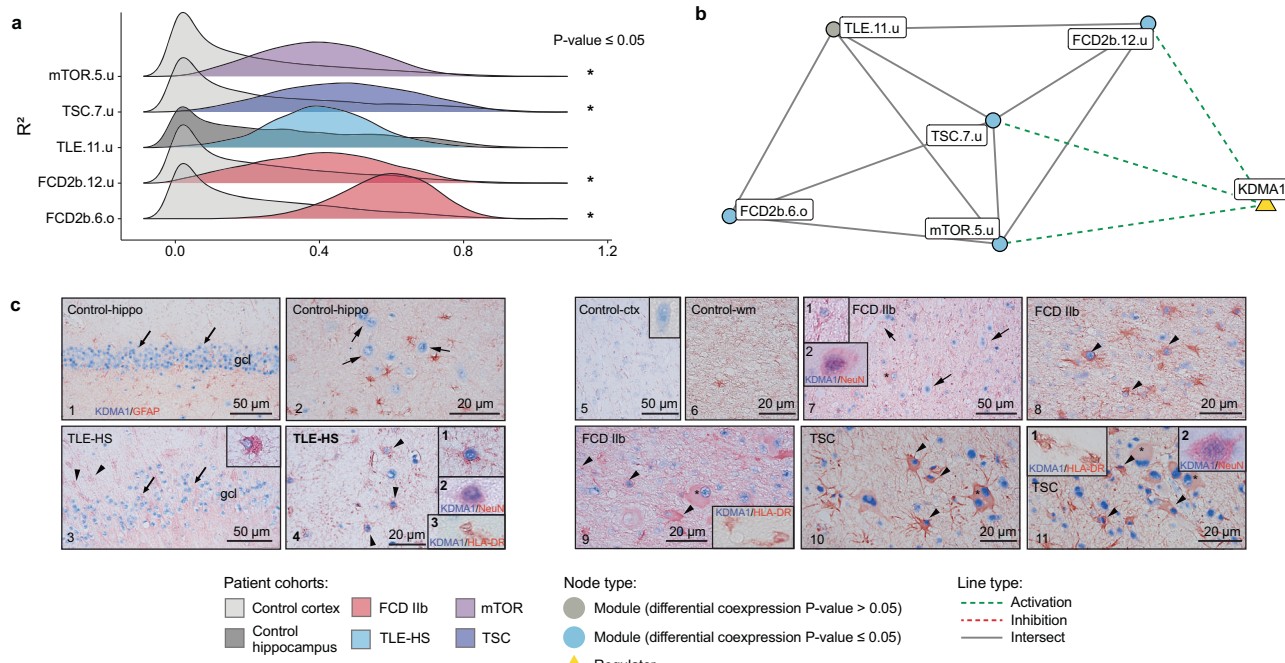

**Fig. 5 | Gene modules differential coexpression for multiple regulomes related to pathological mechanisms.** Network showing the gene overlap size between different gene modules and upstream transcriptional regulators. Cellular expression pattern of KDM1A immunoreactivity (IR) assessed in TLE-HS, FCD IIb, and TSC ($n = 3$ biological replicates per cohorts, $n = 2$ technical replicates). **a** The ridgeplots showed the distribution of gene modules coexpression ($R^2$) for epilepsy and control cohorts within the energy metabolism regulome. Statistical significance of differential coexpression was assessed using a two-sided permutation test (TSC.10.u $p$-value = $3.9 \times 10^{-2}$, FCD2b.7.u $p$-value = $1.46 \times 10^{-2}$). **b** Energy metabolism network highlighting the differentially coexpressed gene modules. KMD1A/LSD1 was predicted as common transcriptional regulator showing activation effect on FCD2b.12.u, TSC.7.u, and mTOR.5.u. **c** Cellular expression of KDM1A IR in TLE-HS, FCD IIb, and TSC. Panels 1–11: IHC of KDM1A. Panels 1–2: In control hippocampus, KDM1A expression was restricted to neuronal cells; KDM1A was not detectable in GFAP-positive cells (astrocytes); Panel 1: Nuclear expression in granule cell layer (gcl; arrows) of the dentate gyrus (DG); Panel 2: Nuclear expression in hilar neurons (arrows). Panels 3–4: In TLE-HS, KDM1A nuclear expression in both neurons (arrows) and astroglial cells (arrowheads). KDM1A expression in a NeuN positive neuron (insert in 2 in panel 4). Absence of KDM1A expression in HLA-DR positive cells (microglia/macrophages; insert 3 in panel 4). Panels 5–6: In control cortex, KDM1A expression was restricted to neuronal cells (insert in panel 5: high-magnification of a positive neuron); KDM1A was not detectable in GFAP-positive cells. Panels 7–9: In FCD IIb, KDM1A IR was observed in dysplastic neurons (arrows) and GFAP-positive cells (arrowheads; insert 1 in panel 7), including GFAP-positive balloon cells (asterisk). KDM1A expression in a NeuN positive dysplastic neuron (insert 2 in panel 7). Absence of KDM1A expression in HLA-DR positive cells (microglia/macrophages; panel 9). Panels 10–11: In TSC, KDM1A IR was observed in dysplastic neurons (arrows) and GFAP-positive cells (arrowheads), including giant cells (asterisks). Absence of KDM1A expression in HLA-DR positive cells (microglia/macrophages; insert 1 in panel 11). KDM1A expression in a NeuN dysplastic neuron (insert 2 in panel 11). Scale bars: 50 μm. FCD focal cortical dysplasia, GFAP glial fibrillary acidic protein, HLA human leukocyte antigen, TLE-HS temporal lobe epilepsy with hippocampal sclerosis, TSC tuberous sclerosis complex. Source data are provided as a Source Data file.

fetal astrocytes (treatment at 3 h and 6 h). The pathway analysis of FCD2b12.u, TSC.7.u, and mTOR.5.u revealed not only impairment of cell metabolism pathways including mitochondria electron transport chain, response to oxidative stress, oxidoreductase complex signaling, ATPase activity, and cellular respiration but also inflammatory response pathways including IL-1 mediated signaling pathways, NF-Kb signaling, T and B cells receptor signaling pathways further demonstrating the tight interplay between energy metabolism and inflammation in epilepsy. Further details of the enriched pathways are reported in Supplementary Data 4. Thus, we aimed at exploring the impact of KDM1A downregulation not only on cellular metabolism, via the expression of ROS markers and cellular ROS production, but also on inflammation. Our in vitro experiment revealed, *KDM1A* was downregulated after *KDM1A* siRNA inhibition in both control and PMA/Ionomycin stimulated cells (3 h/6 h) (Supplementary Fig. 2a). Furthermore PMA/Ionomycin stimulation was confirmed by the upregulation of *MMP3* and *MMP9* (Supplementary Fig. 2b). Finally, *KDM1A* siRNA inhibition showed a significant upregulation of IL1b after 3 h of PMA/Ionomycin stimulation but no change in *C3* expression (Supplementary Fig. 2c). Furthermore, *KDM1A* downregulation showed no impact on the expression of other ROS markers (Supplementary Fig. 2d) and the production of cellular ROS (Supplementary Fig. 2e).

## Neurotransmission and synaptic plasticity
A second enhanced regulome captured neurotransmission and synaptic plasticity showing enrichment for nicotine signaling (Meta-Base), transmission across chemical synapse (Reactome), and chemical synaptic transmission (GO) (Supplementary Data 4 and 5). Cell-type marker enrichment from PanglaoDB identified significant overlap with markers from interneurons and neurons (Supplementary Data 4 and 5). These neurotransmission and synaptic plasticity-related modules showed a differentiated effect across the different pathologies with a significant increase in gene coexpression in FCD IIb (FCD2b.7.u) and TSC (TSC.10.u) (Supplementary Fig. 3a). However, the modules are conserved in both control and epilepsy cohorts. Common upstream regulators NRSF and CoREST have been identified as having an inhibitory effect (Supplementary Fig. 3b). In addition, Supplementary Data 6–8 shows the differential expression results for genes belonging to GABA (Supplementary Data 6) and glutamate receptor (Supplementary Data 7) signaling pathways across the different cohorts and the expression profile of *KCC1* and *KCC2* (Supplementary Data 8).

## Discussion
Chronic DREs are highly heterogeneous but despite differences in etiology and clinical presentations, TLE-HS and mTORopathies (FCD II

and TSC) potentially share downstream molecular mechanisms underlying drug resistance. To our knowledge, this study applies a network-based approach across human epilepsies and independently identify multiple dysregulated biological processes. Upstream regulators identified by CRAFT open the possibility of assessing their ability to restore gene expression towards the healthy state. Multiple studies have provided a proof of principle, demonstrating the modulation of gene networks may restore their homeostatic gene expression profile in the field of epilepsy and neuro-oncology[10,20].

In this study, a global comparison of the transcriptional profile of 162 human brain samples showed separation according to disease and tissue of origin. However, as the epilepsy condition is partly defined by the brain region of seizure origin, the effect of tissue type and disease could not be assessed independently. A more detailed assessment of mTORopathies aligned with well-described histopathological evidence indicates a spectrum from FCD IIa to FCD IIb to TSC cortical tubers. The only discriminator between FCD IIa and FCD IIb is the presence of balloon cells in FCD IIb, which appear to act as crucial drivers of inflammation in this FCD subtype[21]. The low reassignment rate of FCD IIb and TSC cortical tubers may reflect their similar histopathology (balloon cells closely resemble giant cells in TSC) and cell signaling abnormalities[15,21]. The molecular resemblance between FCD IIa, FCD IIb, and TSC patient samples supported the creation of an additional meta-cohort in order to identify transcriptional similarities in the downstream analyses.

To build a regulatory molecular model of the pathobiology, gene modules were identified per cohort. The application of this network-based system analysis, developed by Srivastava et al. [10], revealed different numbers of affected gene modules across the cohorts, in line with the underlying heterogenicity and structure of the population. No association to seizure frequency could be identified in any of the cohorts, suggesting that regulomes may capture the current regulatory networks mostly involved in the pathobiology but not directly affected by seizure frequency. Finally, functional annotation is missing for some modules due to absence of cell-type and pathway enrichment, limiting our current understanding of these pathologies. Furthermore, we acknowledge that the choice of the control material is difficult in human studies, particularly in case of pathologies affecting young patients, which limits the number of cases suitable for gene expression studies. We had the privilege to obtain the human postmortem from young controls. Future work could explore opportunities to incorporate also subjects with non-lesional epilepsy potentially offering a more comprehensive insight into the shared mechanisms unrelated to lesions.

Connecting these identified mechanisms across the DREs enabled a global understanding of disease dysregulations captured by 29 regulomes. Using different metrics, their link to disease biology was established, classifying them as constitutive if present in healthy controls and patients, enhanced regulomes if showing an increased activity in epilepsy, activated regulomes when only present in epilepsy, and finally pathology-specific regulomes. The annotation of these impaired mechanisms identified a diverse array of functions related to immune response, neurotransmission and synaptic plasticity, brain ECM, neuroinflammation, neuronal support and myelination, and energy metabolism, among others. Here, we have focused on more mechanisms identified in the disease state only.

In the TLE-HS patient population, a specific regulome enriched for microglial cell-type markers and associated with immune response and neuroinflammation was identified in module TLE.20.o. Although the relevance of these pathways is not only limited to TLE-HS, this particular gene set was found only to be coregulated in TLE-HS. The activation and function of microglia in combination with upregulation of proinflammatory cytokines and innate immune response receptors are described in TLE-HS patients and status epilepticus (SE)[22]. Srivastava et al. [10]. highlighted the dysregulated neuroinflammatory modules in

pilocarpine mouse model, describing the association to seizure frequency, the conservation in human TLE brain, and the therapeutic efficacy of targeting the predicted regulator, Csf1r. TLE.20.o was shown to correspond to the microglial modules identified in the pilocarpine mouse model (MmPIL.16.o, MmPIL.18.o, MmPil.24.o) based human/mouse gene orthologs[10]. Finally, Csf1R is also predicted as a regulator for TLE.20.o, supporting the robustness and importance of this impaired mechanism in TLE-HS disease pathobiology. The gene modules and correspondence across patient data and animal models enable the construction of a translational disease framework and identification of relevant animal models for subsequent validation.

The mTORopathies presented a specific activated regulome associated with neuronal support and myelination. Multiple studies have shown a link between hyperactivation of mTOR pathway and myelin deficiency, impairment of proliferation and differentiation of oligodendrocytes progenitor cells as well as oligodendroglial turnover[23,24]. Our transcriptomic data corroborate the reported literature findings. CRAFT identified SOX10, a TF essential for the differentiation of myelinating Schwann cells and oligodendrocytes[25], implicated in demyelinating diseases[26]. In addition, miR-488-5p was predicted to inhibit oligodendrocyte-dysregulated modules, however, limited literature is available on the role of this microRNA in the brain[27,28].

The overall comparison of gene modules across epilepsies highlighted the activated regulome related to brain ECM organization and enriched for astrocytes cell-type markers. The brain ECM provides structural and functional support to glia and neurons. Several studies have reported the involvement of astrocytes in different epilepsy models showing SE-induced glial cell death and subsequent enhanced proliferation of immature astrocytes. Modified expression of multiple ECM components affects neurotransmission, synaptic plasticity, and remyelination in the epileptic zone[29]. Seizure activity has been associated with degradation of ECM components and regulators[30] while targeting specific matrix metalloproteinases (MMPs) can reduce seizure severity and frequency in a rat model of TLE[31]. The activity of SP1, the CRAFT predicted regulator, was linked to MMPs in oncology and it was also associated to multiple cellular processes via ECM degradation[32,33]. Recent molecular studies showed that SP1 plays a role in epilepsy, neuronal injury, and maintenance of spontaneous seizure activity[34]. The cellular expression pattern of SP1 IR was confirmed in astroglial cells in TLE-HS as well as dysplastic neurons, astrocytes, and balloon/giant cells across mTORopathy cohorts. The IR in control tissues was sporadic, further supporting SP1 potential role in ECM in epilepsy.

Another activated regulome was identified related to energy metabolism. Different studies observed deficiencies in key components of the glycolytic metabolism and oxidative phosphorylation (OXPHOS), potentially due to oxidative stress, slowing the tricarboxylic acid cycle in epilepsy[35], leading to neuronal hyperexcitability[36] and generation of reactive oxygen species and/or NOX[36]. Our results showed that the (dys)regulation(s) of energy metabolism, was not conserved in healthy tissue, but only became activated in epileptic conditions. Furthermore, the energy metabolism regulome displayed enrichment in multiple pathways. These pathways included those related to both innate and adaptive immune responses, along with the mitochondria electron transport chain, response to oxidative stress, signaling of the oxidoreductase complex, ATPase activity, and cellular respiration. These data further corroborate the interplay between energy metabolism, oxidative stress, and inflammation in epilepsy as ROS are an intrinsic byproduct of ATP production leading to the activation of key proinflammatory molecules triggering a positive feedback loop[37–39]. Multiple studies have demonstrated astrocytes play a critical role in regulating metabolism and redox signaling as well as neuroinflammation[40]. Astrocytes rely on their strong antioxidant capacity and glycolytic handling to provide

metabolic and redox precursors in their cross-talk with neurons[41,42]. CRAFT identified KDM1A (LSD1), which has been reported to modulate OXPHOS in metabolic tissues by genome-wide binding and transcriptome analyses. In addition, an imbalance in KDM1A/neuroKDM1A, a neuron-specific alternative splicing of exon 8a, has been identified to affect neurotransmission, synaptic plasticity[43,44] and hyperexcitability in the pilocarpine mouse model[45]. NeuroKDM1A null mice showed clear reduction in number of seizures and longer latency to first seizure after pilocarpine treatment[45]. KDM1A was predicted to activate the energy metabolism regulome, and although lacking specific cell-type enrichment, its cellular expression pattern in TLE-HS, FCD IIb, and TSC consistently manifested in astrocytes and neurons. Existing literature predominantly explores the role of KDM1A in a neuronal context, prompting a more comprehensive examination of the underlying molecular mechanisms of KDM1A activity in astrocytes. Furthermore, considering the significance of astrocytes in ROS production and immune response, the potential involvement of KDM1A in astrocyte function was also considered[46]. Given the existing body of research on KDM1A in neurons, our focus aimed to investigate its role in an alternative cell type. In our in vitro study the downregulation of KDM1A in PMA/Ionomycin stimulated fetal astrocytes showed increased inflammatory signature upon inhibition whilst no effects could be appreciated on the expression of ROS markers and cellular ROS production. KDM1A plays a role in regulating gene expression by removing specific methyl groups from lysine residues on histone proteins. The role of KDM1A in inflammation is complex, as it can have both proinflammatory and anti-inflammatory effects depending on the context, cell type, and specific molecular pathways involved[47,48]. The dual nature of KDM1A's involvement in inflammation highlights the intricate and context-dependent nature of its functions[49,50]. In line with its inflammatory dual nature, KDM1A role in regulating energy metabolism is controversial. Detectable ROS levels were produced as a byproduct of KDM1A chromatin remodeling activity in osteosarcoma cell lines[51]. In addition, KDM1A increased oxidative stress and ferroptosis promoting renal ischemia and reperfusion injury through activation of TLR4/NOX4 pathway in mice[52]. However, while multiple studies showed KDM1A pro-oxidative stress effect, KDM1A beneficial anti-obesity effects, skeletal muscle regeneration, and the ability of acting as a metabolic sensor for nutritional regulation of metabolic health were reported[53]. Although our results were in line with the literature, exploring the complexity of KDM1A nature in a simplistic model, like the stimulated primary astrocytes in culture, limits the possibility of understanding the underlying molecular mechanisms of KDM1A. Nevertheless, these findings support further investigation into the role of KDM1A in the pathobiology of DRE to determine its therapeutic potential in more complex systems is required.

Finally, altered regulomes related to neuronal function were also identified. Previous studies have already described alterations in neurotransmission related to the balance of excitation/inhibition and immaturity hypothesis link to GABAergic dysfunction in mTORopathies[54,55]. In this study we see similar alteration of the GABAergic and glutamergic signaling. The shared regulome captured is more broadly related to neurotransmission and synaptic plasticity and shows a differentiated effect across the different pathologies with a significant increase in gene coexpression in FCD IIb and TSC further supporting the alteration of neuronal signaling in mTOR-related pathologies[56].

In this study, gene modules were used to establish a computational framework of the epilepsy pathobiology (Fig. 6). We summarize these impaired biological mechanisms as the molecular hallmarks of epilepsy derived from transcriptional profiles and supported by our current understanding of epilepsy pathobiology (Fig. 7). This overview captures the immune response and neuroinflammation regulome enhanced in all epilepsy cohorts and is pathology-specific in TLE-HS as well as the mTORopathy pathology-specific regulome involved in neuronal support and myelination. The brain ECM and energy metabolism regulomes activated across all epilepsy cohorts and the neurotransmission and synaptic plasticity regulome were enhanced in all epilepsy cohorts.

In this study, gene modules were used to describe the molecular heterogenicity of DREs. This network-based system analysis revealed multiple dysregulated coexpression modules in the disease state. Employing the CRAFT framework allowed identification of multiple biological regulators that can be used to assess the therapeutic effect of a module's activity. The systematic comparison across TLE-HS, FCD IIa, FCD IIb, and TSC allowed the identification of impaired mechanisms related to neurotransmission and synaptic plasticity, immune response and neuroinflammation, brain ECM, energy metabolism, and neuronal support and myelination. We propose that these impaired pathways may affect epilepsy development across the studied pathologies, becoming the potential hallmarks of DREs, with the identified upstream protein offering opportunities for drug-target discovery and development.

## Methods

### Patients

Four distinct epilepsy pathologies were considered in this study, namely TLE-HS, FCD IIa, FCD IIb, and TSC cortical tubers. In addition, age- and tissue-matched control tissue samples were collected (control cortex $n = 14$; control hippocampus $n = 13$). Upon patient consent and in accordance to the local ethics committee of the contributing medical centers (science committee of the BioBank and Medical Ethical Committee, Amsterdam UMC - protocol number: 21-174), brain tissues included in this study were obtained from the archives of the Departments of Neuropathology of the Amsterdam UMC (Amsterdam, The Netherlands) and the UMC Utrecht (Utrecht, The Netherlands) (Supplementary Data 9). In addition, all procedures received prior approval by the local ethics committee of the contributing medical centers (science committee of the BioBank and Medical Ethical Committee, Amsterdam UMC - protocol number: 21-174), and were conducted in accordance with the guidelines for good laboratory practice of the European Commission and in accordance with the Declaration of Helsinki and the Amsterdam UMC Research Code. Cortical and hippocampal brain samples were obtained from patients undergoing surgery for intractable epilepsy and diagnosed with FCD type II ($n = 17$ FCD IIa, $n = 33$ FCD IIb), TSC cortical tubers ($n = 21$), and TLE-HS ($n = 64$), respectively (Table 2; more details in Supplementary Data 9).

All cases were reviewed independently by two neuropathologists (A.E. and A.M.). Patients who underwent implantation of strip and/or grid electrodes for chronic subdural invasive monitoring before resection and patients who underwent previous resective epilepsy surgery were excluded from this study. The classification of hippocampal sclerosis (HS) was based on analysis of microscopic examination as described by the International League Against Epilepsy[6]. The diagnosis of FCD was confirmed according to the international consensus classification system proposed for grading FCD[9]. All patients with cortical tubers fulfilled the diagnostic criteria for TSC cortical tubers (including genetic analysis for the detection of germline mutations)[57]. All FCD type II samples underwent deep sequencing using DNA extracted from snap-frozen surgical brain tissue targeting 13 genes (FCD panel SoVarGen, South Korea); analysis for replicated data was performed in accordance with a previous study[58] (Supplementary Data 10). All the cases with a confirmed histological diagnosis of FCD type 2 (both those with detected mutations and those without) and all TSC cases (a germline mutations have been identified in all TSC cases) were included in the mTORopathy cohort.

Control material was obtained at autopsy from age- and brain area-matched control samples that were obtained at autopsy from individuals without a history of seizures or other neurological disease (Table 2; more details in Supplementary Data 9). The causes of death

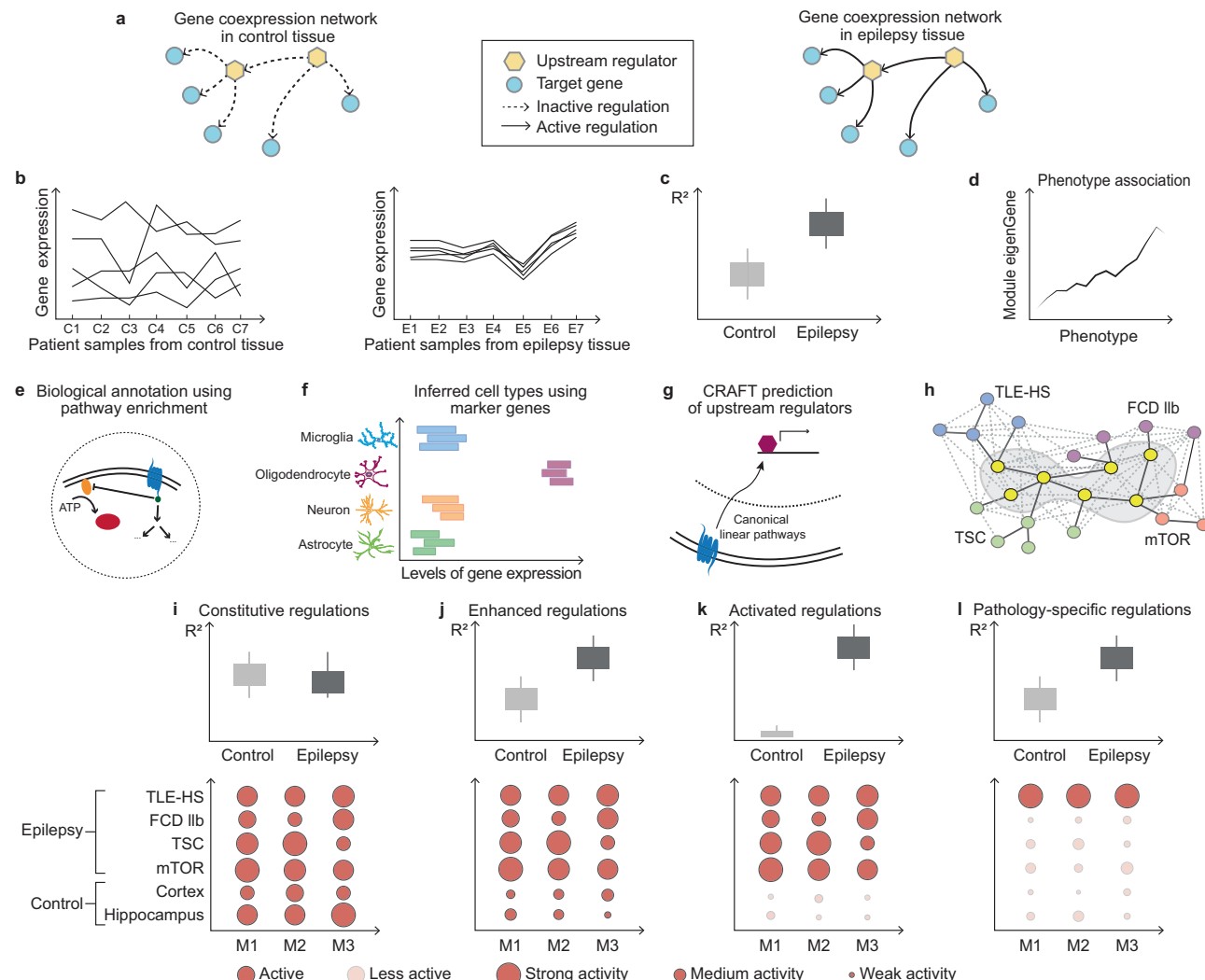

**Fig. 6 | The workflow of gene module annotation and identification of regulomes epilepsies, leading to a proposed summary of impaired biological mechanisms as the molecular hallmarks of drug-resistant epilepsy. a** Gene modules capture the underlying regulatory processes that are present in the disease state. **b** Correlation matrix across the different samples within one cohort. To infer potential biological function, responsible cell type(s), and the link to disease, the following metrics were considered for each gene module: **c** differential coexpression between control and epilepsy ($R^2$), **d** association to phenotype, **e** functional pathway annotation, **f** inferred cell type, and **g** prediction of direct (transcription factor and microRNA) and indirect (cell membrane receptor) upstream regulators. **h** Unsupervised hierarchical clustering identified corresponding clusters of gene modules, termed regulomes. For all regulomes, differential coexpression and conservation were obtained to classify the following four classes of regulations: **i** Constitutive regulations capture those that are present in control and epilepsy patient samples. **j** Enhanced regulations are present in control samples but show enhanced activity in epilepsy patient samples. **k** Activated regulations can only be identified in epilepsy patient samples and may represent strong disease impaired pathways. **l** Some gene modules did not show a strong overlap with gene modules of other epilepsy cohorts while showing significant increase in coexpression in the original epilepsy cohort and were referred to as pathology-specific regulations. ADP adenosine diphosphate, ATP adenosine triphosphate, C1-7 samples from control tissue, CRAFT Causal Reasoning Analytical Framework for Target discovery, E1-7 samples from epilepsy patient tissue, FDC IIb focal cortical dysplasia type IIb, M1-3 gene modules, mTOR mechanistic target of rapamycin, mTORopathies mTOR-related malformations of cortical development, TLE-HS temporal lobe epilepsy with hippocampal sclerosis, TSC tuberous sclerosis complex. Source data are provided as a Source Data file.

for these controls included arrhythmia, myocardial infarction, and acute cardiorespiratory failure. All autopsies were conducted within 12 h after death. Brain tissue was frozen and kept at −80 °C (for molecular analysis) or fixed in 4% paraformaldehyde and embedded in paraffin (FFPE) for histological analysis. All procedures received prior approval by the local ethics committee of the contributing medical centers, and were conducted in accordance with the guidelines for good laboratory practice of the European Commission.

### RNA isolation
For RNA isolation, human tissue was homogenized in 700 μl Qiazol Lysis Reagent (Qiagen Benelux, Venlo, The Netherlands). Total RNA including the microRNA (miRNA) fraction was isolated using the miRNeasy Mini Kit (Qiagen Benelux, Venlo, The Netherlands) according to the manufacturer's instructions. The concentration and purity of RNA was determined at 260/280 nm using a Nanodrop spectrophotometer (Ocean Optics, Dunedin, FL, USA) and RNA integrity was assessed using a Bioanalyser 2100 (Agilent Technologies, Santa Clara, CA, USA). Only samples with RNA integrity number (RIN) equal or greater than 6.0 were used for sequencing.

### RNA-seq library preparation and sequencing
All library preparation and sequencing were performed by GenomeScan (Leiden, The Netherlands). The NEBNext Ultra II Directional RNA Library Prep Kit for Illumina (New England Biolabs, Ipswich, MA, USA) was used to process the samples. Sample preparation was

# Molecular hallmarks of drug-resistant epilepsy

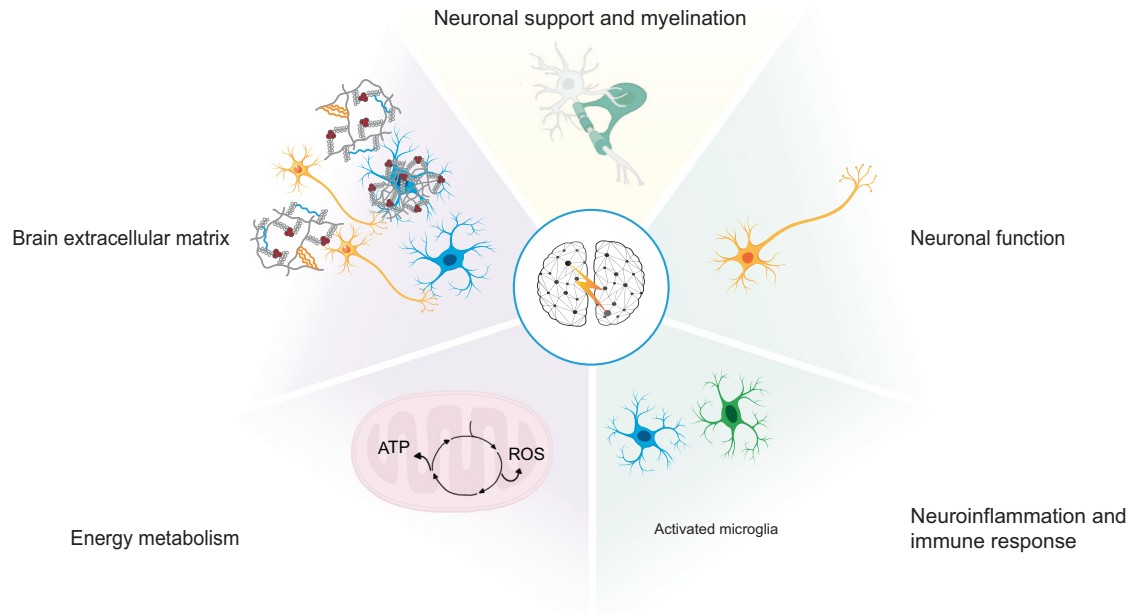

**Fig. 7 | Proposed summary of impaired biological mechanisms as the molecular hallmarks of drug-resistant epilepsy.** This workflow led to a proposal for the molecular hallmarks of drug-resistant epilepsy. Enhanced regulations were identified related to neuronal function and neuroinflammation and immune response. Two activated regulomes were identified and involved in brain extracellular matrix and energy metabolism (oxidative phosphorylation/respiratory electron

transport). Finally, connecting gene coexpression modules across epilepsy cohorts allows the identification of regulations specific to epilepsy cohorts such as neuroinflammation and immune response in TLE-HS and neuronal support and myelination in mTORopathies. mTORopathies mTOR-related malformations of cortical development, TLE-HS temporal lobe epilepsy with hippocampal sclerosis.

**Table 2 | Summary of clinical information of the study cohorts (control cortex, control hippocampus, TLE-HS, FCD IIa, FCD IIb, and TSC cortical tubers)**

|  | Mean age at onset of epilepsy (years) | Mean age surgery (years) | Average seizure frequency (months) | Mutation | | | | | | Medications | | |
|---|---|---|---|---|---|---|---|---|---|---|---|---|
|  |  |  |  | *DEPDC5* | *AKT3* | *MTOR* | *NLPR2/ NLPR3* | *TSC1* | *TSC2* | 1 | 2 | ≥3 |
| Control Cortex (*n* = 14) |  | 21 (0–61) |  |  |  |  |  |  |  |  |  |  |
| Control Hippocampus (*n* = 13) |  | 47 (0–82) |  |  |  |  |  |  |  |  |  |  |
| TLE-HS (*n* = 64) | 12 (0–48) | 35 (2–62) | 24 |  |  |  |  |  |  | 13 | 32 | 19 |
| FCD IIa (*n* = 17) | 5 (0–22) | 11 (0–34) | 356 | 4 | 3 | 4 | 2 |  |  | 1 | 3 | 13 |
| FCD IIb (*n* = 33) | 4 (0–21) | 15 (2–46) | 208 |  |  | 10 |  | 1 |  | 4 | 11 | 18 |
| TSC cortical tubers (*n* = 21) | 3 (0–26) | 7 (0–30) | 148 |  |  |  |  | 6 | 15 | 3 | 6 | 12 |

For detailed information please refer to Supplementary Data 9.

*FCD* focal cortical dysplasia, *TLE-HS* temporal lobe epilepsy with hippocampal sclerosis, *TSC* tuberous sclerosis complex.

performed according to the protocol NEBNext Ultra II Directional RNA Library prep Kit for Illumina (NEB #E7760S/L). Briefly, mRNA was isolated from total RNA using oligo-dT magnetic beads. After fragmentation of mRNA, cDNA synthesis was performed. Next, sequencing adapters were ligated to the cDNA fragments followed by polymerase chain reaction amplification. Clustering and DNA-sequencing was performed using the NovaSeq6000 (Illumina, Foster City, CA, USA) in accordance with manufacturers' guidelines. All samples underwent

paired-end sequencing of 150 nucleotides in length; the mean read depth per sample was 47 million reads.

The Decontamination Using Kmers (BBDuk) tool from the BBTools suite was used for adapter removal, quality trimming and removal of contaminant sequences (ribosomal or bacterial)[59]. A phred33 score of 20 was used to assess the quality of the read, with any read shorter than 31 nucleotides in length excluded from the downstream analysis.

Reads were aligned directly to the human GRCh38 reference transcriptome (Gencode version 33)[60] using Salmon v0.11.3[61]. Transcript counts were summarized to the gene level and scaled using library size and average transcript length using the R package tximport[62]. Genes detected in less than 20% of the samples in any diagnosis cohort and with counts less than six across all samples were filtered out, resulting in 28,366 genes for downstream analysis. The gene counts were then normalized using the weighted trimmed mean of M-values method with the R package edgeR[63]. The normalized counts were then $log_2$ transformed using the voom function from the R package limma[64]. In addition, few significantly upregulated and downregulated genes were selected to validate the results of the RNA-seq analysis from the same cohorts. The confirmatory RT-PCR results of the expression of *CD163* (F: GACAGCGGCTTGCAGTTTC; R: TCTTAA AGGCTGAACTCACTGGG), *CCL3* (F: TGCAACCAGTTCTCTGCATC; R: T GGCTGCTCGTCTCAAAGTA), *CCL2* (F: CCCAAAGAAGCTGTGATCTT CA; R: TCTGGGGAAAGCTAGGGGAA), *IL1b* (F: GCATCCAGCTACGAAT CTCC; R: GAACCAGCATCTTCCTCAGC), and *WNT7B* (F: CCCTCCCT GGATCATGCAC; R: GATGACAGTGCTCCGAGCTT) can be found in Supplementary Fig. 4 and the relative primer sequences in Supplementary Data 11. Furthermore, Boer et al.[65] performed PCR validation of several differential expressed genes in the present study in an independent TSC cohort[65].

## Unsupervised hierarchical clustering and discriminant analysis on principal components

Unsupervised hierarchical clustering based on principal components was used to identify underlying structure in the gene expression matrix using the stats and ggdendro R package[66]. Next, a discriminant analysis of principal components (DAPC) was performed using optim.a.score to identify the optimal number of principal components to retain as implemented by the adegenet R package[66,67].

## Identification of gene coexpression modules

Coexpression networks were constructed per epilepsy cohort using hierarchical clustering of normalized gene expression as developed by Srivastava et al.[10]. First, as healthy matching control samples were age-matched across the general sample set, the age distribution was assessed per cohort before applying the workflow. In addition, any outliers due to area of resection or library preparation were removed. Next, only genes showing high variability across samples were retained (median absolute deviation [MAD] ≥ 0.25). For all remaining genes, the 1-Spearman rank correlation was computed for all gene pairs[68–70] and used to construct the adjacency matrix (soft-thresholding power = 6)[71]. Unsupervised hierarchical clustering using Ward's method identified the clusters of genes[72] (from $K = 1–200$). The optimal number ($K_x$) was obtained by the inflection point of the curve which is calculated based on the second derivative of percentage of the variance explained ($R^2$) per $K$[55]. Next, a leave-one-out bootstrapping procedure was implemented to assess the effect of samples on the stability and robustness of gene coregulation modules. For each permutation, gene coexpression modules were identified using the above-mentioned workflow and records of gene module membership. Cluster membership was used to construct the similarity matrix to identify genes assigned to the junk module based on an arbitrary threshold (50% assigned to junk module). The remaining genes were clustered based on the similarity matrix to obtain the coexpression modules. Finally, the modules were divided using (anti-)correlation of genes within the module. Based on the relative over- or underexpression of the module's genes compared with healthy control samples, each submodule was assigned an o or u suffix, respectively.

To ensure the robustness of the identified modules, coexpression modules were only assembled in epilepsy cohorts with greater than 20 samples. An additional joint analysis was performed across all mTORopathies (FCD IIa, FCD IIb, and TSC cortical tubers). The presence of outliers related to technical covariates was assessed using principal component analysis regression and removed from further analyses.

## Differential coexpression

For each module the correlation between gene expression was calculated in both healthy controls and epilepsy patients to obtain the difference in median $R^2$. The empirical $P$-value was estimated for each module by comparing the difference in median $R^2$ to the null distribution generated by performing 10,000 permutations of samples across cohorts[10,73].

## Phenotype association to module eigenGene

The relationship between module expression and the different reported phenotypes was explored using a linear model between each module's eigenGene and the covariate using lme4 R package: hippocampal sclerosis (HS) subtype, $log_{10}$ of self-reported seizure frequency, gender, age, duration, antiseizure medications, sequencing group, and library preparation batch. As duration also depends on the age of the patients, age was made an additional covariate when assessing association with duration.

## Functional annotation using enrichment analysis

The modules were functionally annotated using multiple pathway resources (MetaCore - Cortellis solution, 11/03/2022, © 2023 Clarivate), Reactome, and GO as well as cell-type enrichment based on marker gene signatures derived from PanglaoDB[74]. A hypergeometric test was used to assess the significance of enriched pathway terms or marker gene signatures using a false discovery rate (FDR) correction to rectify for multiple testing using all expressed genes as a background[75].

## CRAFT framework: in silico causal reasoning

Candidate upstream regulators for the identified gene coexpression modules were predicted using the CRAFT framework. Srivastava et al.[10]. defined a causal reasoning framework that utilizes the direction of effects between the three components of the system, namely CMPs, TFs, and target genes. The interactions between these three components and the direction of these interactions were obtained from the MetaCore (Cortellis solution, 11/03/2022, © 2023 Clarivate), an integrated knowledgebase for pathway analysis of high throughput transcriptomic data. It contains ca. 1600 protein interaction pathways, which are a comprehensive resource of human, mouse, and rat signaling, metabolism, diseases, and stem cells, all manually curated from peer-reviewed literature. The upstream regulator prediction workflow have been developed by Srivastava et al.[10]. All expressed membrane receptors, TFs and target genes from MetaBase were identified. Next, for each TF the set of target genes was retrieved as well as its activity (activation, inhibition, unspecified) and upstream membrane receptors affecting a TF and their effect were obtained using MetaBase® defined canonical linear pathways. The overall effect of the membrane receptor on the underlying module was defined by combining the separate effects of CMP-TF and TF-gene. The significance of effect of a regulator (TF or CMP) on a module was subsequently assessed by testing the overlap between genes under the control of the regulator and the genes belonging to a module (hypergeometric test), taking all expressed genes as the universe. FDR was calculated using Benjamini–Hochberg correction of enrichment P values, taking into account the total number of enrichment tests performed in testing[75].

## Identification of shared epilepsy regulations based on gene coexpression modules

The subsequent paragraph details the identification of specific epilepsy regulations as captured by gene coexpression modules in the independent epilepsy cohorts. Although different structural epilepsies are studied, similar pathways or mechanisms may still be dysregulated.

To identify shared epilepsy regulations, the amount of gene content overlap between the gene coexpression modules from each epilepsy cohorts was identified using the inclusion index:

$$\text{inclusion index} = \frac{\text{length(intersect}(x,y))}{\min(\text{length}(x),\text{length}(y))} \quad (1)$$

with $x$ and $y$ as two gene coexpression modules. Next, unsupervised hierarchical clustering based on Ward's method was used to identify modules that showed overlap in gene content[72] using the silhouette method to identify the optimal number of clusters. The analyses were performed with the stats and factoextra R packages[76]. By design, within an epilepsy cohort, a gene can only belong to one coexpression module. Therefore, the intersect between gene coexpression modules across epilepsy cohorts was defined as those genes occurring in at least one module per epilepsy cohort. This gene intersection was subsequently submitted to a hypergeometric test to obtain functional annotation with pathway resources (MetaBase, Reactome, GO) as well as cell-type enrichment based on marker gene signatures derived from PanglaoDB[74]. Finally, the conservation of gene coexpression in other epilepsy cohorts and healthy control tissue was assessed with the same permutation approach as for differential coexpression analysis.

## Immunohistochemistry

Human brain tissue was fixed in 10% buffered formalin and embedded in paraffin. Paraffin-embedded tissue was sectioned at 6 µm, mounted on pre-coated glass slides (Star Frost, Waldemar Knittel Glasbearbeitungs, Braunschweig, Germany), and processed for immunohistochemical staining ($n = 3$ biological replicates per cohorts, $n = 2$ technical replicates). Immunohistochemistry was carried out on samples from patients as reported in Supplementary Data 9. The following protocol was used as previously described[21]: the following antibodies and dilutions were applied: SOX10 (SOX10, rabbit monoclonal, Cell Marque, 383R-16, EP268, Lot#: 0000209147, 1:200) incubated 1 h at room temperature (RT), SP1 (SP1, rabbit monoclonal, Abcam, EPR6662(B), ab124804, Lot#: GR3281146-2, 1:200) and KMD1A/lysine-specific demethylase 1 (LSD1) (KMD1A/LSD1, rabbit polyclonal, Cell Signaling Technology, Cat#: 2139S, Lot#: 2, 1:200) incubated overnight at 4 °C. For double labeling of SOX10, SP1 and KMD1A/LSD1, sections were incubated with NeuN (NeuN, mouse monoclonal, clone A60, MAB377; Chemicon, Temecula, CA, USA; 1:2000), glial fibrillary acidic protein (GFAP; mouse monoclonal, clone GA5, MAB360, Sigma-Aldrich, St. Louis, MO, USA; 1:4000) and HLA-DP/DR/DQ (HLA-II, mouse monoclonal, clone CR3/43, M0775, Agilent Technologies, Santa Clara, CA, USA; 1:100) antibodies, after incubation with the primary antibodies overnight at 4 °C. For detection, sections were first incubated with Brightvision poly-alkaline phosphatase-anti-rabbit (DVPR55AP, Immunologic, Duiven, The Netherlands) for 30 min at room temperature, and washed with phosphate-buffered saline and then with Tris−HCl buffer (0.1 M, pH 8.2) to adjust the pH. Alkaline phosphatase activity was visualized with the alkaline phosphatase substrate kit III Vector Blue (SK-5300, Vector Laboratories Inc., CA, USA). After washing in phosphate-buffered saline, sections were secondly incubated with Brightvision poly-horseradish peroxidase-anti-mouse (DPVM55HRP, Immunologic, Duiven, The Netherlands) for 30 min at room temperature. Signal was detected using the chromogen 3-amino-9-ethylcarbazole (AEC, Sigma- Aldrich, St. Louis, MO, USA) in 0.05 M acetate buffer filtered substrate solution. Sections incubated without the primary antibodies or with the primary antibodies followed by heating treatment were essentially blank.

## Functional assessment in PMA/Ionomycin stimulated fetal astrocytes

Astrocytes were isolated using a papain dissection method (Worthington Biochemical, Lakewood, NJ, USA) from human control brain tissue derived from abortions occurred between gestational weeks 12 and 16. All tissue was collected with written consent and according to the declaration of Helsinki as well as the Amsterdam research code of the medical ethics committee (science committee of the BioBank and Medical Ethical Committee, Amsterdam UMC - protocol number: 21-174). Upon isolation, fetal astrocytes were cultured in DMEM/F10 (1:1) (Gibco, Life Technologies, Grand Island, NY, USA) supplemented with 50 units/ml penicillin and 50 µg/ml streptomycin (1% P/S), 1% L-Glutamine and 10% fetal calf serum (FCS; Gibco, Life Technologies, Grand Island, NY, USA). All cultures were grown and maintained in a 5% $CO_2$ incubator at 37 °C. For experiments, cells ($n = 3$ biological replicates, $n = 2$ technical replicates) were seeded in 12-well plates with $0.1 \times 10^6$ cells/well and allowed to adhere for 48 h. After 48 h, cells were transfected with KDM1A silencer (siRNA id: 108658, Catalog #: AM16708, Interrogated Sequence (Refseq): NM_001009999.2 and NM_015013.3, Thermo Fisher Scientific, Wilmington, DE, USA). Oligonucleotides were delivered to the cells using Lipofectamine® RNAiMax Transfection Reagent (Invitrogen™, Catalog #: 13778075) in a final concentration of 12.5 pmol for a total of 24 h for mRNA isolation. Data of KDM1A siRNA transfected cells were normalized to the control group. This control group consisted of cells exposed to Silencer® Select Negative control N1 siRNA (siRNA id: 4390843, Catalog #: 4390843, Thermo Fisher Scientific, Wilmington, DE, USA), data are expressed as a fold-change compared to the control group.

For RNA isolation, human tissue was homogenized in 700 µl Qiazol Lysis Reagent (Qiagen Benelux, Venlo, The Netherlands). Total RNA including the microRNA (miRNA) fraction was isolated using the miRNeasy Mini Kit (Qiagen Benelux, Venlo, The Netherlands) according to the manufacturer's instructions. The concentration and purity of RNA was determined at 260/280 nm using a Nanodrop spectrophotometer (Ocean Optics, Dunedin, FL, USA) and RNA integrity was assessed using a Bioanalyser 2100 (Agilent Technologies, Santa Clara, CA, USA).

For the evaluation of mRNA expression, qPCRs targeting of *KDM1A* (F: ACCGCCCTATGCAAGGAATA; R: CGCTTCCAACTCCTGAA GTTTT), *C3* (F: CCTGAAGATAGAGGGTGACCA; R: CCACCACGTCCCAG ATCTTA), *IL1b* (F: GCATCCAGCTACGAATCTCC; R: GAACCAGCATC TTCCTCAGC), *MMP3* (F: CTCCAACCGTGAGGGAAAATC; R: CATGGA ATTTCTCTTCTCATCAAA), *MMP9* (F: GAACCAATCTCACCGACAGG; R: GCCACCCGAGTGTAACCATA), were run with *EIF1-a* (F: ATC-CACCTTTGGGTCGCTTT; R: CCGCAACTGTCTGTCTCATATCAC) and *C1orf43* (F: GATTTCCCTGGGTTTCCAGT; R: ATTCGACTCTCCAGGG TTCA) as a housekeeping genes. Quantification of data was performed using the computer program LinRegPCR in which linear regression on the Log (fluorescence) per cycle number data is applied to determine the amplification efficiency per sample[77]. For the relative expression, all groups were compared to the controls. The sequences of the selected primers can be found in Supplementary Data 11.

The cellular determination of ROS was performed using CellROX® Green Reagents (C10444, Thermo Fisher Scientific, Wilmington, DE, USA). The cells were transfected with KDM1A siRNA and treated with PMA/Ion at 3 h and 6 h. The CellROX® Reagent was added to the medium at the end of incubation time at a final concentration of 5 µM to the cells and further incubated for 30 min at 37 °C. Media was removed and the cells were washed three times with PBS. Fluorescent intensity was measured with a Clariostar plate reader (BMG Labtech).

## Reporting summary

Further information on research design is available in the Nature Portfolio Reporting Summary linked to this article.

## Data availability

The data generated in this study are provided in the Supplementary Information/Source Data file. The data generated in this study are available through the Gene Expression Omnibus at https://www.ncbi.nlm.nih.gov/geo with accession number GSE256068. Source data are provided with this paper.

## Code availability

The code used in this study are deposited in the GitHub repository.

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

## Acknowledgements

The authors would like to acknowledge Ciara Duffy, PhD, CMPP (Envision Pharma Group, Sydney, Australia) for editorial support, which was funded by UCB Pharma. Publication coordination was provided by Tom Grant, PhD (UCB Pharma, Slough, UK). We would like to thank Alessandra Tammaro, Valentin Jean-Pierre, Eric Jnoff, Baptiste Manteau, Rodrigo Aguirre, Isabelle Niespodziany, Matthew Page, and Christian Wolff their support and feedback on the research and writing of this manuscript. This study was funded by UCB Pharma. E.A. received funding from The Netherlands Organisation for Health Research and Development (ZonMw) and the European Union's Horizon 2020 research and innovation program under grant agreement No 952455 (EpiNet).

## Author contributions

E.A. and A.M. helped with the selection and collection and revision of human brain tissues and clinical data. L.F. and A.R. performed analysis of RNA sequencing data. A.R., M.J.L., and J.J.A. performed the experiments and immunohistochemistry. P.G., L.F., and A.S. developed and improved

the methodology. E.A., S.D., J.D.M., M.R., J.v.E., and P.G. conceived the study and participated in its design and coordination. L.F. and A.R. drafted and prepared the manuscript. All authors read, revised, and approved the final manuscript.

## Competing interests

A.R., E.A., and J.D.M. received an unrestricted grant from UCB Pharma. L.F., P.G., A.S., M.R., J.v. E., and S.D. are employees of UCB Pharma, and P.G., J.v. E., A.S., and S.D. receive stock or stock options from their employment. The remaining authors declare no competing interests.
