## [Peer Review File · Nature Communications]

Identification of gene regulatory networks affected across drug-resistant epilepsiesREVIEWER COMMENTS

Reviewer #1 (Remarks to the Author):

Epileptic pathology is highly heterogeneous depending on the region of origin in the brain and exhibits a spectrum of disorders. In this manuscript Lisbeth and colleagues have made a good effort in understanding commonalities among different forms of epilepsy using a system biology approach. The authors start by generating transcriptome data from large cohort of human patients with four distinct epilepsy pathologies including temporal lobe epilepsy with hippocampal sclerosis and mTORopathies. They then employ unsupervised clustering of cohorts as well as supervised discriminant analysis of various principal components in these cohorts. Previously published Causal reasoning framework (CRAFT), was eventually used to identify upstream regulators and to identify various modules or regulomes based on differential gene co-expression, compared to control tissue data. The authors identify several shared key regulatory gene networks common to drug resistant epilepsies. Using a systematic GO analysis of these modules/regulomes they identify pathways in immune response, myelination, extracellular matrix modulation, energy metabolism and neurotransmission. They go on to validate expression of two potential regulatory genes SP1 and LSD1 by Immunohistochemistry in tissues from patients. Observations and findings made in this manuscript are in line with previously known pathways that are mis-regulated during epilepsy, but interestingly across different forms of epilepsy. The manuscript provides a potentially valuable resource from a large cohort, varying in age, brain region and epilepsy pathology.

Overall, the manuscript is well organized and written but I have a few comments and suggestions, mostly minor modifications in figures and text that might help convey the message better. It would also be good to see a few more validations of the findings made here, especially if the authors have access to diseased tissue samples.

1) Fig.1: All the sub-figures of this panel show the transcriptional difference between cohorts based on the origin of tissue and the type of epilepsy. The first dendrogram is not very informative, and colored bars are too compressed making it difficult visually. All the figures here could be modified or condensed to fewer panels but convey the same message. Perhaps including a heatmap in addition to/replacing one of them might be helpful.

A few more details of what is represented (scores/values) in discriminant analysis (panel d and e) might be helpful.

2) Please restructure lines 116 to 118, under results section: "Identification of gene co-expression modules within epilepsy pathologies".

3) Fig.3 and table 1 and text in the corresponding seem to have a few mismatching details in number of modules or number of significant modules. For e.g., TLE-HS, 9 gene modules are significantly changed but table 1, shows it as 9. Similarly, for TSC and mTORopathies the numbers do not match.

4) Please confirm the number of regulomes identified and analyzed/discussed correspond in Line 204, line 325 in discussion and referred to in supplementary table 5 /Fig.2b.

5) There are several instances where the reader might wish for proper reference or pointer towards a figure or table that make it a better read.

6) Although there are quite a lot of interesting observations from this systematic approach it would be good to focus on one or two regulated pathways and validate them further.

Reviewer #2 (Remarks to the Author):

This is a very interesting article, which uses a technique called "CRAFT" (Causal Reasoning Analytical Framework for Target discovery) to explore disease transcriptional profiles across several different pathologies that produce the epilepsy phenotype, including temporal lobe epilepsy with hippocampal sclerosis (TLE-HS) and malformations of cortical development, including focal cortical dysplasia type IIa and type IIb (FCD IIa and FCD IIb) and cortical tubers in tuberous sclerosis complex(TSC).

Several pathways were identified that were activated or present only in either one or several of the studied pathologies. Many of these pathways are already on the radar screen for therapy development, but this will provide additional impetus to address these pathways, and possibly focus therapeutic targets to specific pathologies.

A strength of the study is that pathologies were confirmed independently by two neuropathologists. A minor weakness is that no subjects were included who did not have an epileptic lesion, as that might have been elucidating for non-lesion related shared mechanisms. Although many surgeries are done after localization of the focus around a lesion, there are many patients who have electrical localization of non-lesional foci.

One other question is the impact of anti-seizure medication on any of these pathways, since all patients were taking 1-3 anti-seizure medications at the time of surgery.

Reviewer #3 (Remarks to the Author):

In the current study, Francois et al. performed RNA-seq of brain tissues obtained from refractory epilepsy patients (TLS-HS, FCD2a, FCD2b, TSC) and compared their gene expression profile with post-mortem brain specimens (cortex and hippocampus). They performed gene co-expression analysis to identify modules enriched in the cohort. Further, they perform differential gene co-expression analysis to identify and delineate epilepsy- and pathology-specific modules. The modules were functionally annotated for the pathway and cell type enrichment. They used CRAFT to identify upstream CMP/TFs/miRNA potentially driving expression/differential expression of the identified modules in the cohort. Overall, the study involves robust bioinformatics. The approach described in the study could prove useful in identifying drug targets besides providing meaningful insights into the molecular mechanisms of drug resistant epilepsy.

However the authors need to address the following queries:

Major queries:

1. Several studies on the transcriptome analysis of DREs are already reported. The authors have not referred to some of the papers published on transcriptome analysis of brain tissues resected from patients with MTLHS and DNET. (Dixit et al., *Genomics*, 2016, Kumar et al., *Funct Integr Genomics*. 2022). The authors should discuss all the papers reporting transcriptome analysis of various DREs.
2. The authors are performing transcriptome network-based analysis not across epilepsies, rather drug resistant epilepsies. The authors should also discuss about the possibility of modulation of the gene networks due to the AEDs as the control samples are post-mortem samples without AEDs.
3. How the control sample was collected (sample collection time after death) and checked for integrity? The clinical data of the control samples such as cause of death etc is not mentioned.
4. The authors should also discuss about various underlying mechanisms proposed so far specifically in FCD like dysmaturity hypotheisis (Cepeda et al., *Epilepsy Behav*. 2006) and if they find any correlation with the existing hypotheses. Previous studies suggested altered glutamatergic activity in TLE-HS (Banerjee et al., *Scientific Reports*, 2017) and altered GABAergic activity in FCD (Banerjee et al., *Front Cell Neurosci*, 2020) could they find any correlation with such altered synaptic activity in the two DRE pathologies.
5. Since the authors are proposing SOX10 and miR-488-5p as potential molecules also supported by literature as a marker of hyperactivation of mTOR pathway and myelin deficiency, impairment of proliferation and differentiation of oligodendrocytes progenitor cells. Their expression levels should have been confirmed by other experiments such as IHC or realtime PCR. The authors have performed IHC for SP1 and LSD1 and both of them are upregulated. It is always preferable to select few molecule showing significant upregulation and few down regulation as

a confirmatory technique to correlate with the RNAseq analysis.

6. The authors claim that the identified affected gene module and regulators may provide novel opportunities to modulate these networks and restore their homeostatic gene expression profile. They need to provide some supporting data in literature suggesting how modulation of gene networks can restore homeostatic gene expression profile.

Minor queries:

1. How was the cut-off of gene co-expression (R2) determined?
2. check line 646-648/Table 2: Were only FCD II samples subjected to 13 gene FCD panel deep sequencing for identifying mTor mutations?
3. Were all FCDIIa, IIb, and TSC samples included in the mTORopathy cohort or only those with the identified mutations?
4. RIN values for the samples are not mentioned.

REVIEWER COMMENTS

Reviewer #1 (Remarks to the Author):

Epileptic pathology is highly heterogenous depending on the region of origin in the brain and exhibits a spectrum of disorders. In this manuscript Lisbeth and colleagues have made a good effort in understanding commonalities among different forms of epilepsy using a system biology approach. The authors start by generating transcriptome data from large cohort of human patients with four distinct epilepsy pathologies including temporal lobe epilepsy with hippocampal sclerosis and mTORopathies. They then employ unsupervised clustering of cohorts as well as supervised discriminant analysis of various principal components in these cohorts. Previously published Causal reasoning framework (CRAFT), was eventually used to identify upstream regulators and to identify various modules or regulomes based on differential gene co-expression, compared to control tissue data. The authors identify several shared key regulatory gene networks common to drug resistant epilepsies. Using a systematic GO analysis of these modules/regulomes they identify pathways in immune response, myelination, extracellular matrix modulation, energy metabolism and neurotransmission. They go on to validate expression of two potential regulatory genes SP1 and LSD1 by Immunohistochemistry in tissues from patients. Observations and findings made in this manuscript are in line with previously known pathways that are mis-regulated during epilepsy, but interestingly across different forms of epilepsy. The manuscript provides a potentially valuable resource from a large cohort, varying in age, brain region and epilepsy pathology.

Overall, the manuscript is well organized and written but I have a few comments and suggestions, mostly minor modifications in figures and text that might help convey the message better. It would also be good to see a few more validations of the findings made here, especially if the authors have access to diseased tissue samples.

Authors reply: We greatly appreciate the reviewer's positive assessment of the manuscript's organization and writing. We acknowledge the importance of enhancing the clarity of our figures and text to effectively communicate our message. We agree with the reviewer's suggestion regarding the inclusion of additional validations for our findings.

1) Fig.1: All the sub-figures of this panel show the transcriptional difference between cohorts based on the origin of tissue and the type of epilepsy. The first dendrogram is not very informative, and colored bars are too compressed making it difficult visually. All the figures here could be modified or condensed to fewer panels but convey the same message. Perhaps including a heatmap in addition to/replacing one of them might be helpful. A few more details of what is represented (scores/values) in discriminant analysis (panel d and e) might be helpful.

Authors reply: We appreciate the feedback on the clarity and presentation of the sub-figures in this panel. The suggestions for improvement are well-received and will greatly contribute to enhancing the visual communication of our results.

Regarding the first dendrogram, we understand the reviewer's concern. We refined this dendrogram to make figure 1A more visually interpretable. Additionally, according to the reviewer's suggestion, we edited the legend of figure 1d and figure 1e (line 603 to 611, page 22) as follows in order to better convey the message of the figures and describe what is represented: "D, Prior and posterior cohort assignment after discriminant analysis on principal components on all cohorts. The prior and posterior assignment of individuals to the cohort based on the discriminant functions is provided indicating admixture between cohorts. The numbers in the heatmap indicate how many samples of each cohort are (re)assigned to the same cohort based on the discriminant functions. E, Prior and posterior cohort assignment after discriminant analysis on principal components on mTORopathies specifically. The prior and posterior assignment of individuals to the cohort based on the discriminant functions were provided indicating admixture between cohorts. The numbers in the heatmap indicate how many samples of each cohort are (re)assigned to the same cohort based on the discriminant functions."

2) Please restructure lines 116 to 118, under results section: "Identification of gene co-expression modules within epilepsy pathologies".

Authors reply: We appreciate the reviewer's suggestion to restructure lines 116 to 118. Taking this advice into consideration, we have rephrased the section, now in line 128 to 130 in page 5, as follows: "Briefly, pathway and cell type annotations aimed to unravel the underlying pathobiology of the diseases. Furthermore, the differential coexpression of gene modules between disease and healthy control samples brought to light the gene modules impacted in the disease state."

3) Fig.3 and table 1 and text in the corresponding seem to have a few mismatching details in number of modules or number of significant modules. For e.g., TLE-HS, 9 gene modules are significantly changed but table 1, shows it as 9. Similarly, for TSC and mTORopathies the numbers do not match.

Authors reply: Thank you for bringing this discrepancy to our attention. We apologize for the inconsistency between the information presented in Fig. 2, Table 1, and the corresponding text. We greatly appreciate the careful review, and we have amended these errors to ensure the accuracy and coherence of our manuscript. Taking this advice into consideration, we have restructured the paragraphs now in lines 144 to 146 in page 5 as follows: "For TLE-HS, 37 gene modules were identified with nine modules presenting a significant change in coexpression as measured by R^2 between disease and healthy control patient samples, indicating that these modules were significantly affected in TLE-HS (Fig. 2a, panel TLE-HS)". Lines 171-172 in page 6 were rephrased as follows: "In TSC, 30 gene modules were identified with 23 gene modules significantly differentially coexpressed (Fig. 2a, panel TSC)." Lastly, lines 184-185 in page 7 were edited as follows: "In the mTOR cohort (all FCD IIa, FCD IIb and TSC samples), 28 gene modules were identified but only nine gene modules were found differentially coexpressed (Fig. 2a, panel mTORopathy)".

4) Please confirm the number of regulomes identified and analyzed/discussed correspond in Line 204, line 325 in discussion and referred to in supplementary table 5 /Fig.2b.

Authors reply: Thank you for raising this important point regarding the consistency of regulomes across different parts of our manuscript. Upon thorough review, we can confirm that the number of regulomes identified and analyzed corresponds as follows:

Lines 220-225, page 8: This section was edited as follows: "The analysis revealed 29 regulomes total varying in size from two to 10 gene modules (Fig. 2b, Supplementary Table 5). Here, regulomes (n = 14) with a consistent functional annotation across multiple pathway databases and effect in epilepsy were identified and selected (Fig. 2b). Based on the classification described above, regulomes related to neurotransmission and synaptic plasticity, immune response, brain ECM, energy metabolism and oligodendrocyte function are highlighted (Fig. 2b)."

Lines 359-360, in page 12 in the Discussion: We have cross-checked and ensured that the number of regulomes discussed aligns with the correct count. The text was edited as follows: "Connecting these identified mechanisms across the DREs enabled a global understanding of disease dysregulations captured by 29 regulomes."

Supplementary Table 5: The number of regulomes referred to in Supplementary Table 5 has been verified to match the accurate count.

Fig. 2b: We have also cross-verified the information in Fig. 2b to ensure consistency.

5) There are several instances where the reader might wish for proper reference or pointer towards a figure or table that make it a better read.

Authors reply: Thank you for the feedback. We completely agree that clear references and pointers towards figures are crucial for guiding readers and ensuring they can easily locate relevant visual aids. In light of the reviewer's suggestion, we thoroughly reviewed the manuscript to identify instances where references to figures or tables are lacking and we highlighted all the new references to the tables and figures in yellow.

6) Although there are quite a lot of interesting observations from this systematic approach it would be good to focus on one or two regulated pathways and validate them further.

Authors reply: We carefully considered the reviewer's suggestion and explored the feasibility of validation by focusing on one regulator and its associated pathways. Specifically, we investigated the functional role of LSD1 to further validate the expression data provided.

Since we observed the upregulation of LSD1 proteins in astrocytes within epilepsy-associated pathologies, we employed primary cultures of human fetal astrocyte to investigate the effects of their *in vitro* manipulation on inflammation and oxidative stress. Details on the experimental design are now provided in the "Materials and Methods" section of the manuscript in line 907 to 933, page 30. The *in vitro* results obtained were incorporated in the "Results" section of the manuscript in lines 306 to 312, page 10 and in the Supplementary Fig. 3. In addition, we included these data in the "Discussion" section of the manuscript, in lines 410 to 421, pages 13 and 14.

Reviewer #2 (Remarks to the Author):

This is a very interesting article, which uses a technique called "CRAFT" (Causal Reasoning Analytical Framework for Target discovery) to explore disease transcriptional profiles across several different pathologies that produce the epilepsy phenotype, including temporal lobe epilepsy with hippocampal sclerosis (TLE-HS) and malformations of cortical development, including focal cortical dysplasia type IIa and type IIb (FCD IIa and FCD IIb) and cortical tubers in tuberous sclerosis complex(TSC).

Several pathways were identified that were activated or present only in either one or several of the studied pathologies. Many of these pathways are already on the radar screen for therapy development, but this will provide additional impetus to address these pathways, and possibly focus therapeutic targets to specific pathologies.

A strength of the study is that pathologies were confirmed independently by two neuropathologists. A minor weakness is that no subjects were included who did not have an epileptic lesion, as that might have been elucidating for non-lesion related shared mechanisms. Although many surgeries are done after localization of the focus around a lesion, there are many patients who have electrical localization of non-lesional foci.

Authors reply: We thank the reviewer for the positive comments and interest in our manuscript. The utilization of the "CRAFT" method has indeed enabled us to identify disease transcriptional profiles across various pathologies linked to the epilepsy phenotype.

Regarding their point about not including subjects without lesions (i.e. non-lesional TLE or perilesional tissue from FCD or TSC cases), we acknowledge the potential significance of elucidating non-lesion-related shared mechanisms (this consideration is now included in the discussion in lines 353 to 358 in page 12). In future work, we will explore opportunities to incorporate subjects with non-lesional foci, which could enhance the comprehensiveness of our analysis and provide a better understanding of the mechanisms involved. Of course, an ideal experimental design we would like to include a sufficient number (at least n= 30) of non-lesional tissue samples to apply the "CRAFT" method. However, not all the patients with non-lesional epilepsy undergo surgical resection and particularly working on rare genetic disorders, one cannot expect to be able to perform this ideal experiment. Moreover, for diagnostic purposes it is important to have paraffin-embedded material, thus frozen material is not always available. In case of FCD and TSC only in a limited number of cases the resection includes a significant amount of peri-lesional (histologically normal) cortex and in the large majority of cases also this material is embedded in paraffin. Moreover, a comparison with non-epileptogenic tubers could not be performed, since this tissue is not resected during epilepsy surgery and infrequently available at autopsy.

One other question is the impact of anti-seizure medication on any of these pathways, since all patients were taking 1-3 anti-seizure medications at the time of surgery.

Authors reply: We sincerely appreciate the emphasis on the importance of considering patients' medication regimens in our analysis. We fully recognize the impact that ASMs can have on our findings. This aspect has been thoroughly integrated into our study's methodology and results.

We conducted a comprehensive correlation analysis to carefully examined the potential relationship between gene expression within each module and the clinical data, while accounting for the influence of ASMs. Notably, our findings revealed no significant association between gene expression and the patient's medication regimen within our cohort. These results are now clearly mentioned in the relevant section of our manuscript.

Materials and method, lines 839-842, in page 33: The relationship between module expression and the different reported phenotypes was explored using a linear model between each module's eigenGene and the covariate: HS subtype, log10 of self-reported seizure frequency, sex, age, duration, antiseizure medications, sequencing group and library preparation batch.

Results, line 139 in page 5: No association to phenotype and antiseizure medications was identified for the modules in any cohort.

Reviewer #3 (Remarks to the Author):

In the current study, Francois et al. performed RNA-seq of brain tissues obtained from refractory epilepsy patients (TLS-HS, FCD2a, FCD2b, TSC) and compared their gene expression profile with post-mortem brain specimens (cortex and hippocampus). They performed gene co-expression analysis to identify modules enriched in the cohort. Further, they perform differential gene co-expression analysis to identify and delineate epilepsy- and pathology-specific modules. The modules were functionally annotated for the pathway and cell type enrichment. They used CRAFT to identify upstream CMP/TFs/miRNA potentially driving expression/differential expression of the identified modules in the cohort. Overall, the study involves robust bioinformatics. The approach described in the study could prove useful in identifying drug targets besides providing meaningful insights into the molecular mechanisms of drug resistant epilepsy.

However the authors need to address the following queries:

Authors reply: We appreciate the reviewer's comprehensive assessment of our study. We are pleased that the reviewer recognizes the rigor of our bioinformatics approach and the potential significance of our findings. We are grateful for the positive feedback and will ensure that the manuscript accurately reflects the suggestions provided below.

Major queries:

1. Several studies on the transcriptome analysis of DREs are already reported. The authors have not referred to some of the papers published on transcriptome analysis of brain tissues resected from patients with MTLHS and DNET. (Dixit et al., Genomics, 2016, Kumar et al., Funct Integr Genomics. 2022). The authors should discuss all the papers reporting transcriptome analysis of various DREs.

Authors reply: We appreciate the reviewers' input and acknowledge the importance of referencing relevant studies on transcriptome analysis in patients with drug-resistant epilepsies. We apologize for not including all the studies due to limitations in the number of references. In light of the reviewer's suggestions, we will incorporate the references to the papers by Dixit et al. (Genomics, 2016) and Kumar et al. (Funct Integr Genomics, 2022) in our discussion. Thank you for bringing this to our attention, and we will ensure a more comprehensive coverage of relevant transcriptome analysis studies in the revised manuscript.

2. The authors are performing transcriptome network-based analysis not across epilepsies, rather drug resistant epilepsies. The authors should also discuss about the possibility of modulation of the gene networks due to the AEDs as the control samples are post-mortem samples without AEDs.

Authors reply:

Thank you for bringing up a crucial point regarding the potential impact of anti-seizure medications (ASMs) on our identified pathways. We completely agree that the medication regimen of patients is an important factor to consider (as pointed out by both reviewer 2 and 3). This point has been already taken in consideration in our analysis. We performed a correlation analysis with the ASM and no association between gene expression, within each module, and the medication regimen of patients. This information is now clear included in the materials and methods and result section.

Materials and method, lines 839-842, in page 33: The relationship between module expression and the different reported phenotypes was explored using a linear model between each module's eigenGene and the covariate: HS subtype, log10 of self-reported seizure frequency, sex, age, duration, antiseizure medications, sequencing group and library preparation batch.

Results, line 139 in page 5: No association to phenotype and antiseizure medications was identified for the modules in any cohort.

3. How the control sample was collected (sample collection time after death) and checked for integrity? The clinical data of the control samples such as cause of death etc is not mentioned.

Authors reply: The hippocampus and cortex of age-matched controls without a history of seizures or other neurological diseases were obtained through autopsy. The causes of death for these controls included arrhythmia, myocardial infarction, and acute cardiorespiratory failure. All autopsies were conducted within 12 hours after death. As detailed in the "Materials and Methods" section, we assessed RNA integrity using a 2100 (Agilent Technologies, Santa Clara, CA, USA), ensuring that only samples with RNA Integrity Number (RIN) equal to or greater than 6.0 were used for sequencing. We have now incorporated this information in the "Materials and methods" in lines 760 to 762 in page 30 and lines 769-770 page 30, for clarity.

4. The authors should also discuss about various underlying mechanisms proposed so far specifically in FCD like dysmaturity hypothesis (Cepeda et al., Epilepsy Behav. 2006) and if they find any correlation with the existing hypotheses. Previous studies suggested altered glutamatergic activity in TLE-HS (Banerjee et al., Scientific Reports, 2017) and altered GABAergic activity in FCD (Banerjee et al., Front Cell Neurosci,2020) could they find any correlation with such altered synaptic activity in the two DRE pathologies.

Authors reply: According to the reviewer's suggestion, we acknowledge the importance of discussing various underlying mechanisms proposed for TLE, FCD, and TSC, including alteration of the balance excitation/inhibition and the "immaturity hypothesis" pointing to a GABAergic dysfunction in mTORopathies. We incorporated a short section in the manuscript's results and discussion in lines 322 to 325 in page 11 and lines 424 to 430 in page 14 respectively, outlining these mechanisms and their potential correlation with our findings. Furthermore, we included three supplementary tables (Supplementary Tables 8-10) indicating the differential expression results for genes belonging to GABA and glutamate receptor signaling pathways across the different cohorts and the expression profile of KCC1 and KCC2.

5. Since the authors are proposing SOX10 and miR-488-5p as potential molecules also supported by literature as a marker of hyperactivation of mTOR pathway and myelin deficiency, impairment of proliferation and differentiation of oligodendrocytes progenitor cells. Their expression levels should have been confirmed by other experiments such as IHC or realtime PCR. The authors have performed IHC for SP1 and LSD1 and both of them are upregulated. It is always preferable to select few molecule showing significant upregulation and few down regulation as a confirmatory technique to correlate with the RNAseq analysis.

Authors reply: Following the reviewer's suggestion, we conducted additional experiments to validate the expression levels of SOX10 (via IHC) and miR-488-5p (using in situ hybridization, ISH) in patient tissues. Unfortunately, attempts to perform in situ hybridization for miR-488 on human FFPE material were unsuccessful

due to technical difficulties encountered with the designed probe. The results of the IHC were incorporated in figure 3c. The legend was amended accordingly in line 632 to 642, page 22.

Additionally, we sought to further validate the expression data by investigating the functional roles of LSD1. Since we observed the upregulation of LSD1 proteins in astrocytes within epilepsy-associated pathologies, we employed primary cultures of human fetal astrocyte to investigate the effects of their *in vitro* manipulation on inflammation and oxidative stress. Details on the experimental design are now provided in the "Materials and Methods" section of the manuscript in lines 911 to 937, page 35-36. The *in vitro* results obtained were incorporated in the "Results" section of the manuscript in lines 306 to 312, page 10 and in the Supplementary Fig. 3. In addition, we included these data in the "Discussion" section of the manuscript, in lines 409 to 421, pages 13 and 14.

According to the reviewer's suggestion, we selected few significantly upregulated and downregulated genes to validate the results of the RNASeq analysis from the same cohort. We indicated the genes the PCR was performed on in the "Materials and methods" section "RNA-Seq library preparation and sequencing", lines 796 to 801 page 26. The confirmatory results can be found in the Supplementary figure 4. Furthermore, Boer et al., 2010 (PMID: 19912235) shows PCR validation of several differential expressed genes in the present study in an independent TSC cohort. This information is now included in the "Materials and methods" section "RNA-Seq library preparation and sequencing", lines 800-801 page 32.

6. The authors claim that the identified affected gene module and regulators may provide novel opportunities to modulate these networks and restore their homeostatic gene expression profile. They need to provide some supporting data in literature suggesting how modulation of gene networks can restore homeostatic gene expression profile.

Authors reply: We appreciate the opportunity to provide more context and evidence for our claims.

In the introduction and discussion sections, we highlighted the utility of the "CRAFT" method in identifying disease-specific regulatory modules. This method was employed using data generated from an experimental model of temporal lobe epilepsy. Through this approach, we were able to identify Csf1R signaling as a regulator of the epileptogenic network. This finding serves as a proof of principle, demonstrating that disease-context-specific effects on epilepsy can indeed be achieved through specific modules predicted by the CRAFT method (PMID: 30177815).

Additionally, we would like to refer to a recent study involving integrated systems-genetic analyses. In this study, a multitarget network characteristic relevant to glioma progression was identified. The study successfully validated one of the predicted targets *in vitro*, thereby confirming the feasibility and efficacy of network-based multitarget drug discovery. This finding supports the notion that modifying pathological networks can have a positive impact, even within the field of neuro-oncology (PMID: 31420939).

These referenced studies (now included in the "Discussion" section, line 332 to 334, page 11) collectively provide additional evidence that supports our assertion that targeting gene networks can indeed lead to the restoration of homeostatic gene expression profiles in disease contexts. We hope that these examples further illustrate the potential of our findings in the context of modulating networks for therapeutic purposes.

Minor queries:

1. How was the cut-off of gene co-expression (R2) determined?

Authors reply: In accordance with the reviewer's request, this information has been amended and made clearer in the "Materials and Methods" section.

Line 817 to 819, page 32: The optimal number (K_x) was obtained by the inflection point of the curve which is calculated based on the second derivative of percentage of the variance explained (R^2) per K .

2. check line 646-648/Table 2: Were only FCD II samples subjected to 13 gene FCD panel deep sequencing for identifying mTor mutations?

Authors reply: We can confirm that only cases with a confirmed histological diagnosis of FCD type II underwent deep sequencing. Therefore, the sentence previously in line 646-648 and now in line 755 to 757, page 30 and Table 2 is indeed accurate.

3. *Were all FCDIIa, IIb, and TSC samples included in the mTORopathy cohort or only those with the identified mutations?*

Authors reply: The mTORopathy cohort includes all cases with a confirmed histological diagnosis of FCD type 2 (both those with detected mutations and those without) and all TSC cases (a germline mutations have been identified in all TSC cases). This information has now been incorporated in the manuscript in the "Materials and methods" section, "Patients" paragraph (line 751 to 753, page 30).

4. *RIN values for the samples are not mentioned.*

Authors reply: As reported in the "Materials and Methods" section ("RNA isolation" paragraph), RNA integrity was assessed using Bioanalyser 2100 (Agilent Technologies, Santa Clara, CA, USA). Only samples with RNA integrity number (RIN) equal or greater than 6.0 were used for sequencing. This information has now been incorporated in the manuscript in line 773-774, in page 31.

REVIEWER COMMENTS

Reviewer #1 (Remarks to the Author):

In the updated draft of the paper, the authors have made significant improvements and have successfully addressed most of my concerns. They have also rectified the inconsistencies and carried out further validations as recommended. These enhancements have strengthened the study and added to the robustness of the paper. Towards validation experiments, the authors have chosen to explore LSD1 that was identified in the regulome capturing energy metabolism. It is also interesting that this regulome was affected only in the epilepsy cohorts. Few recent studies have shown the importance of LSD1 in OXPHOS, making it relevant to the findings of this study. But I have major concerns in the way it has been presented.

The authors' decision to study LSD1 in the context of energy metabolism, yet conducting experiments in stimulated fetal astrocytes in culture, is intriguing. The main text should provide a more comprehensive explanation of the rationale behind these experiments, including the results and discussion. The authors should clarify why they opted to investigate inflammatory responses or stress responses, instead of focusing on OXPHOS or energy metabolism-related readouts. Furthermore, while the discussion primarily centers on the role of neuronal LSD1, it lacks any significant mention of energy metabolism in astrocytes or the relevance of LSD1 in astrocytes.

There is also an absence of pertinent references highlighting the importance of LSD1 in OXPHOS or metabolism. This omission results in a noticeable disconnect between the validation performed and its significance in the context of epilepsy. It is suggested that the authors either articulate these points in their writing or conduct additional experiments with suitable readouts.

Minor concern:

Please check the references made to the figures in the section Energy metabolism (Line 292) in the main text. There seems to be some discrepancies.

Also, changing the figure subsection legends to numerical than alphabetically could help easy reading and better understanding? For example, in Figure 3c (subsections/panels) now labeled a-j to i to x?

Reviewer #2 (Remarks to the Author):

Thank you, my review comments have been addressed

Reviewer #3 (Remarks to the Author):

The authors have incorporated/addressed all the changes and concerns.

Reviewer #3 (Remarks on code availability):

The authors have incorporated/addressed all the changes and concerns.

REVIEWER COMMENTS

Reviewer #1 (Remarks to the Author):

In the updated draft of the paper, the authors have made significant improvements and have successfully addressed most of my concerns. They have also rectified the inconsistencies and carried out further validations as recommended. These enhancements have strengthened the study and added to the robustness of the paper. Towards validation experiments, the authors have chosen to explore LSD1 that was identified in the regulome capturing energy metabolism. It is also interesting that this regulome was affected only in the epilepsy cohorts. Few recent studies have shown the importance of LSD1 in OXPHOS, making it relevant to the findings of this study. But I have major concerns in the way it has been presented.

The authors' decision to study LSD1 in the context of energy metabolism, yet conducting experiments in stimulated fetal astrocytes in culture, is intriguing. The main text should provide a more comprehensive explanation of the rationale behind these experiments, including the results and discussion. The authors should clarify why they opted to investigate inflammatory responses or stress responses, instead of focusing on OXPHOS or energy metabolism-related readouts.

Furthermore, while the discussion primarily centers on the role of neuronal LSD1, it lacks any significant mention of energy metabolism in astrocytes or the relevance of LSD1 in astrocytes.

There is also an absence of pertinent references highlighting the importance of LSD1 in OXPHOS or metabolism. These omission results in a noticeable disconnect between the validation performed and its significance in the context of epilepsy. It is suggested that the authors either articulate these points in their writing or conduct additional experiments with suitable readouts.

Authors reply:

We greatly appreciate the reviewer's positive assessment of the manuscript's organization and writing. We acknowledge the importance of providing a comprehensive explanation of the rationale behind the experiments performed. The immunohistochemistry validation of the expression of LSD1 showed a strong expression of the protein of interest in astrocytes across all the cohorts. We opted to perform experiments in primary cultures of human astrocytes following PMA/ionomycin stimulation to further explore the impact of LSD1 on metabolic changes within astrocytes and all associated aspects. We acknowledge the limitation of this model being a very simplistic representation of the disease environment. LSD1 was predicted to activate the energy metabolism regulome, which, in turn, displayed enrichment in multiple pathways. These pathways included those related to both innate and adaptive immune responses, along with the mitochondria electron transport chain, response to oxidative stress, signaling of the oxidoreductase complex, ATPase activity, and cellular respiration. In line with the reviewer's suggestion regarding the role of LSD1 in OXPHOS, our experimental design now integrates a more thorough investigation of the energy metabolism regulome, exploring its connections with oxidative stress responses and inflammation.

All edits to the text are highlighted in yellow in the current version of the manuscript.

Supplementary Figure 3 was amended including the results of other functional readouts exploring the reactive oxygen species (ROS) markers gene expression and lastly, the cellular ROS production.

The legend of Supplementary Figure 3 was edited as follows:

Lines 686-696, and page 24: “**d**, LSD1 downregulation after 3h of PMA/Ionomycin stimulation did not induce changes in the expression of ROS markers compared to LSD1 downregulation alone. **e**, LSD1 downregulation after 3h of PMA/Ionomycin stimulation did not provide significant changes in cellular ROS production in astrocytes.”

Lines 700-709, and page 24-25: “**f**. The cellular expression pattern of SP1 IR was assessed in TLE-HS, FCD IIb and TSC. **Panels 1-9:** IHC of SP1. **Panels 1-2** In control hippocampus, SP1 expression was very low in neuronal cells (arrow in b, hilar neuron); SP1 was not detectable in GFAP positive cells. **Panels 2-4:** In TLE-HS, SP1 expression in astroglial cells (arrowheads). **Panels 5-6:** In control cortex, very low expression of SP1 (panel e); occasionally few GFAP positive cells were observed in the white matter (wm) (panel f). **Panels 7-8:** In FCD IIb, SP1 IR was observed in dysplastic neurons (arrows) and GFAP positive cells (arrowheads), including GFAP positive balloon cells (asterisks). SP1 expression in a NeuN dysplastic neuron (insert in g). Absence of SP1 expression in HLA-DR positive cells (microglia/macrophages; insert in h). **Panel 9:** In TSC, SP1 expression in dysplastic neurons (arrow; high-magnification of a dysplastic neuron; insert i3) and GFAP positive cells (arrowheads; insert i1), including giant cells (asterisks). Absence of SP1 expression in HLA-DR positive cells (microglia/macrophages; insert i2).”

Lines 712-725, and page 25: **i**, Cellular expression of LSD1 IR in TLE-HS, FCD IIb and TSC. **Panels 1-11:** IHC of LSD1. **Panels 1-2:** In control hippocampus, LSD1 expression was restricted to neuronal cells; LSD1 was not detectable in GFAP positive cells (astrocytes); **Panel 1:** Nuclear expression in granule cell layer (gcl; arrows) of the dentate gyrus (DG); **Panel 2:** Nuclear expression in hilar neurons (arrows). **Panels 3-4:** In TLE-HS, LSD1 nuclear expression in both neurons (arrows) and astroglial cells (arrowheads). LSD1 expression in a NeuN positive neuron (insert d2). Absence of LSD1 expression in HLA-DR positive cells (microglia/macrophages; insert d3). **Panels 5-6:** In control cortex, LSD1 expression was restricted to neuronal cells (insert in e: high-magnification of a positive neuron); LSD1 was not detectable in GFAP positive cells. **Panels 7-9:** In FCD IIb, LSD1 IR was observed in dysplastic neurons (arrows) and GFAP positive cells (arrowheads; insert g1), including GFAP positive balloon cells (asterisk). LSD1 expression in a NeuN positive dysplastic neuron (insert g2). Absence of LSD1 expression in HLA-DR positive cells (microglia/macrophages; panel i). **Panels 10-11:** In TSC, LSD1 IR was observed in dysplastic neurons

(arrows) and GFAP positive cells (arrowheads), including giant cells (asterisks). Absence of LSD1 expression in HLA-DR positive cells (microglia/macrophages; insert k1). LSD1 expression in a NeuN dysplastic neuron (insert k2). FCD, focal cortical dysplasia; GFAP, glial fibrillary acidic protein; HLA, human leukocyte antigen; TLE-HS, temporal lobe epilepsy with hippocampal sclerosis; TSC, tuberous sclerosis complex.

Further, **the Materials and Methods section, in lines 990-995, page 32**, was implemented as follows to incorporate the functional readouts exploring ROS: “The cellular determination of ROS was performed using CellROX® Green Reagents (C10444, Thermo Fisher Scientific, Wilmington, DE, USA). The cells were transfected with LSD1 siRNA and treated with PMA/Ion at 3h and 6h. The CellROX® Reagent was added to the medium at the end of incubation time at a final concentration of 5 µM to the cells and further incubated for 30 minutes at 37°C. Media was removed and the cells were washed three times with PBS. Fluorescent intensity was measured with a Clariostar plate reader (BMG Labtech).”

According to the reviewer’s suggestions we have now implemented the results and the discussion section as follows:

Results “Energy Metabolism” section, Lines 306-322, page 10-11: This section was edited as follows: “As the IHC of epilepsy cohorts showed a consistent expression of LSD1 in astrocytes, *in vitro* validation of the role of LSD1 was assessed using PMA/Ionomycin stimulated fetal astrocytes (treatment at 3h and 6h). The pathway analysis of FCD2b.12.u, TSC.7.u and mTOR.5.u revealed not only impairment of cell metabolism pathways including mitochondria electron transport chain, response to oxidative stress, oxidoreductase complex signalling, ATPase activity and cellular respiration but also inflammatory response pathways including IL1 mediated signaling pathways, NF-Kb signaling, T and B cells receptor signaling pathways further demonstrating the tight interplay between energy metabolism and inflammation in epilepsy. Further details of the enriched pathways are reported in Supplementary Table 4. Thus, we aimed at exploring the impact of LSD1 downregulation not only on cellular metabolism, via the expression of ROS markers and cellular ROS production, but also on inflammation. Our *in vitro* experiment revealed, LSD1 was downregulated after LSD1 siRNA inhibition in both control and PMA/Ionomycin stimulated cells (3h/6h). Furthermore PMA/Ionomycin stimulation was confirmed by the upregulation of MMP3 and MMP9. Finally, LSD1 siRNA inhibition showed a significant upregulation of IL1b after 3h of PMA/Ionomycin stimulation but no change in C3

expression (Supplementary Fig. 3). Furthermore, LSD1 downregulation showed no impact on the expression of other ROS markers and the production of cellular ROS (Supplementary Fig. 3)."

Discussion, Lines 418-427, page 14. This section was edited as follows: "Furthermore, the energy metabolism regulome displayed enrichment in multiple pathways. These pathways included those related to both innate and adaptive immune responses, along with the mitochondria electron transport chain, response to oxidative stress, signaling of the oxidoreductase complex, ATPase activity, and cellular respiration. These data further corroborate the interplay between energy metabolism, oxidative stress and inflammation in epilepsy as ROS are an intrinsic byproduct of ATP production leading to the activation of key proinflammatory molecules triggering a positive feedback loop³⁷⁻³⁹. Multiple studies have demonstrated astrocytes play a critical role in regulating metabolism and redox signaling as well as neuroinflammation⁴⁰. Astrocytes rely on their strong antioxidant capacity and glycolytic handling to provide metabolic and redox precursors in their cross-talk with neurons^{41,42}"

Discussion, Lines 432-442, page 14. This section was edited as follows: "LSD1 was predicted to activate the energy metabolism regulome, and although lacking specific cell type enrichment, its cellular expression pattern in TLE-HS, FCD IIb, and TSC consistently manifested in astrocytes and neurons. Existing literature predominantly explores the role of LSD1 in a neuronal context, prompting a more comprehensive examination of the underlying molecular mechanisms of LSD1 activity in astrocytes. Furthermore, considering the significance of astrocytes in ROS production and immune response, the potential involvement of LSD1 in astrocyte function was also considered⁴⁶. Given the existing body of research on LSD1 in neurons, our focus aimed to investigate its role in an alternative cell type. In our *in vitro* study the downregulation of LSD1 in PMA/Ionomycin stimulated fetal astrocytes showed increased inflammatory signature upon inhibition whilst no effects could be appreciated on the expression of ROS markers and cellular ROS production."

Discussion, Lines 447-458, page 15. This section was edited as follows: "In line with its inflammatory dual nature, LSD1 role in regulating energy metabolism is controversial. Detectable ROS levels were produced as a byproduct of LSD1 chromatin remodeling activity in osteosarcoma cell lines⁵¹. In addition, LSD1 increased oxidative stress and ferroptosis promoting renal ischemia and reperfusion injury through activation of TLR4/NOX4 pathway in mice⁵². However, while multiple studies showed LSD1 pro-oxidative stress effect, LSD1 beneficial anti-obesity effects, skeletal muscle regeneration and the ability of acting as a metabolic sensor for nutritional regulation of metabolic health were reported⁵³. Although our results were in line with the literature, exploring the complexity of LSD1 nature in a simplistic model, like the stimulated primary astrocytes in culture, limits the possibility of

understanding the underlying molecular mechanisms of LSD1. Nevertheless, these findings support further investigation into the role of LSD1 in the pathobiology of DRE to determine its therapeutic potential in more complex systems is required.”

Finally, we express our appreciation for the reviewer's input and recognize the importance of referencing all relevant studies in support of our findings. We apologize for omitting references related to the role of LSD1 in astrocytes. In response to the reviewer's suggestions, we have incorporated additional references into our discussion and particularly reference 37-39, 46, 51-53. We have now ensured a more comprehensive coverage of the role of LSD1 in both energy metabolism and astrocytes in the revised manuscript.

Minor concern:

Please check the references made to the figures in the section Energy metabolism (Line 292) in the main text. There seems to be some discrepancies.

Also, changing the figure subsection legends to numerical than alphabetically could help easy reading and better understanding? For example, in Figure 3c (subsections/panels) now labeled a-j to i to x?

Authors reply: Thank you for bringing this discrepancy to our attention. We apologize for the inconsistency between the information presented in the section on “Energy Metabolism” in line 292 in the main text and the corresponding figures. We greatly appreciate the careful review, and we have amended these errors to ensure the accuracy and coherence of our manuscript.

Furthermore, we appreciate the feedback on the clarity and presentation of the sub-figures in the panels. The suggestions for improvement are well received and will greatly contribute to enhancing the clear and more comprehensive communication of our results.

We agree with the reviewer's suggestion regarding the use of numbers instead of letters in the subsections of the different panels. We have now amended the panels in Fig 3 and the corresponding legends.

REVIEWERS' COMMENTS

Reviewer #1 (Remarks to the Author):

Thanks to the authors for accommodating my suggestions in the current version of the manuscript. The authors have also addressed all my concerns. In the current version authors have clearly explained the rationale behind the experiments conducted, included additional experiments and discussed the importance of interactions between Neurons and non-neuronal cells like astrocytes in epilepsy.

REVIEWERS' COMMENTS

Reviewer #1 (Remarks to the Author):

Thanks to the authors for accommodating my suggestions in the current version of the manuscript. The authors have also addressed all my concerns. In the current version authors have clearly explained the rationale behind the experiments conducted, included additional experiments and discussed the importance of interactions between Neurons and non-neuronal cells like astrocytes in epilepsy.

Authors reply:

We greatly appreciate the reviewer's constructive revision on the research analyses, experimental work and structure of the discussion concerning the interactions with neurons and glial cells in context of epilepsy.